# The Broad Optimality of Profile Maximum Likelihood

**Yi Hao**
Dept. of Electrical and Computer Engineering
University of California, San Diego
yih179@ucsd.edu

**Alon Orlitsky**
Dept. of Electrical and Computer Engineering
University of California, San Diego
alon@ucsd.edu

## Abstract

We study three fundamental statistical learning problems: distribution estimation, property estimation, and property testing. We establish the profile maximum likelihood (PML) estimator as the first unified sample-optimal approach to a wide range of learning tasks. In particular, for every alphabet size $k$ and desired accuracy $\varepsilon$:

**Distribution estimation** Under $\ell_1$ distance, PML yields optimal $\Theta(k/(\varepsilon^2 \log k))$ sample complexity for sorted distribution estimation, and a PML-based estimator empirically outperforms the Good-Turing estimator on the actual distribution;

**Additive property estimation** For a broad class of additive properties, the PML plug-in estimator uses just four times the sample size required by the best estimator to achieve roughly twice its error, with exponentially higher confidence;

**$\alpha$-Rényi entropy estimation** For an integer $\alpha > 1$, the PML plug-in estimator has optimal $k^{1-1/\alpha}$ sample complexity; for non-integer $\alpha > 3/4$, the PML plug-in estimator has sample complexity lower than the state of the art;

**Identity testing** In testing whether an unknown distribution is equal to or at least $\varepsilon$ far from a given distribution in $\ell_1$ distance, a PML-based tester achieves the optimal sample complexity up to logarithmic factors of $k$.

With minor modifications, most of these results also hold for a near-linear-time computable variant of PML.

## 1 Introduction

A distribution $p$ over a discrete alphabet $\mathcal{X}$ of size $k$ corresponds to an element of the simplex

$$\Delta_{\mathcal{X}} := \left\{ p \in \mathbb{R}^k_{\geq 0} : \sum_{x \in \mathcal{X}} p(x) = 1 \right\}.$$

A distribution *property* is a mapping $f : \Delta_{\mathcal{X}} \to \mathbb{R}$ associating a real value with each distribution. A distribution property $f$ is *symmetric* if it is invariant under domain-symbol permutations. A symmetric property is *additive*, i.e., additively separable, if it can be written as $f(p) := \sum_x f(p(x))$, where for simplicity we use $f$ to denote both the property and the corresponding real function.

Many important symmetric properties are additive. For example,

- **Support size** $S(p) := \sum_x \mathbb{1}_{p(x)>0}$, a fundamental quantity arising in the study of vocabulary size [29, 53, 67], population estimation [34, 52], and database studies [37].

- **Support coverage** $C_m(p) := \sum_x (1 - (1 - p(x))^m)$, where $m$ is a given parameter, the expected number of distinct elements observed in a sample of size $m$, arising in biological [17, 49] and ecological [17–19, 23] research;

- **Shannon entropy** $H(p) := -\sum_x p(x) \log p(x)$, the primary measure of information [24, 66] with numerous applications to machine learning [14, 22, 63] and neuroscience [30, 51];

- **Distance to uniformity** $D(p) := \|p - p_u\|_1$, where $p_u$ is the uniform distribution over $\Delta_{\mathcal{X}}$, a property being central to the field of distribution property testing [10, 12, 15, 65].

Besides being additive and symmetric, these four properties have yet another attribute in common. Under the appropriate interpretation, they are also all 1-Lipschitz. Specifically, for two distributions $p, q \in \Delta_{\mathcal{X}}$, let $\Gamma_{p,q}$ be the collection of distributions over $\mathcal{X} \times \mathcal{X}$ with marginals $p$ and $q$ on the first and second factors respectively. The *relative earth-mover distance* [70], between $p$ and $q$ is

$$R(p,q) := \inf_{\gamma \in \Gamma_{p,q}} \mathbb{E}_{(X,Y) \sim \gamma} \left| \log \frac{p(X)}{q(Y)} \right|.$$

One can verify [70, 71] that $H$, $D$, and $\tilde{C}_m := C_m/m$ are all 1-Lipschitz on the metric space $(\Delta_{\mathcal{X}}, R)$, and $\tilde{S} := S/k$ is 1-Lipschitz over $(\Delta_{\geq 1/k}, R)$, the set of distributions in $\Delta_{\mathcal{X}}$ whose nonzero probabilities are at least $1/k$. We will study all such Lipschitz properties in later sections.

An important symmetric non-additive property is *Rényi entropy*, a well-known measure of randomness with numerous applications to unsupervised learning [44, 77] and image registration [50, 54]. For a distribution $p \in \Delta_{\mathcal{X}}$ and a non-negative real parameter $\alpha \neq 1$, the $\alpha$-*Rényi entropy* [64] of $p$ is $H_\alpha(p) := (1 - \alpha)^{-1} \log \left( \sum_x p_x^\alpha \right)$. In particular, denoted by $H_1(p) := \lim_{\alpha \to 1} H_\alpha(p)$, the 1-*Rényi entropy* is exactly Shannon entropy [64].

## 1.1 Problems of interest

In this work, we consider three fundamental statistical learning problems concerning the estimation and testing of distributions and their properties.

### (Sorted) distribution estimation

A natural learning problem is to estimate an unknown distribution $p \in \Delta_{\mathcal{X}}$ from an i.i.d. sample $X^n \sim p$. For any two distributions $p, q \in \Delta_{\mathcal{X}}$, let $\ell(p,q)$ be the *loss* when we approximate $p$ by $q$. A *distribution estimator* $\hat{p} : \mathcal{X}^* \to \Delta_{\mathcal{X}}$ associates every sequence $x^n \in \mathcal{X}^*$ with a distribution $\hat{p}(x^n)$. We measure the performance of an estimator by its *sample complexity*

$$n(\hat{p}, \varepsilon, \delta) := \min\{n : \forall p \in \Delta_{\mathcal{X}}, \Pr_{X^n \sim p} (\ell(p, \hat{p}(X^n)) \geq \varepsilon) \leq \delta\},$$

the smallest sample size that $\hat{p}$ requires to estimate all distributions in $\Delta_{\mathcal{X}}$ to a desired accuracy $\varepsilon > 0$, with error probability $\delta \in (0, 1)$. The sample complexity of distribution estimation over $\Delta_{\mathcal{X}}$ is

$$n(\varepsilon, \delta) := \min\{n(\hat{p}, \varepsilon, \delta) : \hat{p} : \mathcal{X}^* \to \Delta_{\mathcal{X}}\},$$

the lowest sample complexity of any estimator. For simplicity, we will omit $\delta$ when $\delta = 1/3$.

For a distribution $p \in \Delta_{\mathcal{X}}$, we denote by $\{p\}$ the multiset of its probabilities. The *sorted $\ell_1$ distance* between two distributions $p, q \in \Delta_{\mathcal{X}}$ is

$$\ell_1^<(p,q) := \min_{p' \in \Delta_{\mathcal{X}} : \{p'\} = \{p\}} \|p' - q\|_1 ,$$

the smallest $\ell_1$ distance between $q$ and any sorted version of $p$. As illustrated in Section 7.1 of the supplementary material, this is essentially the 1-Wasserstein distance between uniform measures on the probability multisets $\{p\}$ and $\{q\}$. We will consider both the sorted and unsorted $\ell_1$ distances.

### Property estimation

Often we would like to estimate a given property $f$ of an unknown distribution $p \in \Delta_{\mathcal{X}}$ based on a sample $X^n \sim p$. A *property estimator* is a mapping $\hat{f} : \mathcal{X}^* \to \mathbb{R}$. Analogously, the *sample complexity* of $\hat{f}$ in estimating $f$ over a set $\mathcal{P} \subset \Delta_{\mathcal{X}}$ is

$$n_f(\hat{f}, \mathcal{P}, \varepsilon, \delta) := \min\{n : \forall p \in \mathcal{P}, \Pr_{X^n \sim p} (|\hat{f}(X^n) - f(p)| \geq \varepsilon) \leq \delta\},$$

the smallest sample size that $\hat{f}$ requires to estimate $f$ with accuracy $\varepsilon$ and confidence $1 - \delta$, for all distributions in $\mathcal{P}$. The sample complexity of estimating $f$ over $\mathcal{P}$ is

$$n_f(\mathcal{P}, \varepsilon, \delta) := \min\{n_f(\hat{f}, \mathcal{P}, \varepsilon, \delta) : \hat{f} : \mathcal{X}^* \to \mathbb{R}\},$$

the lowest sample complexity of any estimator. For simplicity, we will omit $\mathcal{P}$ when $\mathcal{P} = \Delta_{\mathcal{X}}$, and omit $\delta$ when $\delta = 1/3$. The standard "median trick" shows that $\log(1/\delta) \cdot n_f(\mathcal{P}, \varepsilon) \geq \Omega(n_f(\mathcal{P}, \varepsilon, \delta))$. By convention, we say an estimator $\hat{f}$ is *sample-optimal* if $n_f(\hat{f}, \mathcal{P}, \varepsilon) = \Theta(n_f(\mathcal{P}, \varepsilon))$.

**Property testing: Identity testing**

A closely related problem is distribution property testing, of which identity testing is the most fundamental and well-studied [15, 32]. Given an error parameter $\varepsilon$, a distribution $q$, and a sample $X^n$ from an unknown distribution $p$, *identity testing* aims to distinguish between the null hypothesis

$$H_0 : p = q$$

and the alternative hypothesis

$$H_1 : \|p - q\|_1 \geq \varepsilon.$$

A *property tester* is a mapping $\hat{t} : \mathcal{X}^* \to \{0, 1\}$, indicating whether $H_0$ or $H_1$ is accepted. Analogous to the two formulations above, the *sample complexity* of $\hat{t}$ is

$$n_q(\hat{t}, \varepsilon, \delta) := \min\{n \colon \forall i \in \{0, 1\} \text{ and } \forall p \in H_i, \Pr_{X^n \sim p} \left( \hat{t}(X^n) \neq i \right) \leq \delta\},$$

and the sample complexity of identity testing with respect to $q$ is

$$n_q(\varepsilon, \delta) := \min\{n(\hat{t}, \varepsilon, \delta) \colon \hat{t} : \mathcal{X}^* \to \{0, 1\}\}.$$

Again, when $\delta = 1/3$, we will omit $\delta$. For $q = p_u$, the problem is also known as *uniformity testing*.

## 1.2 Profile maximum likelihood

The *multiplicity* of a symbol $x \in \mathcal{X}$ in a sequence $x^n := x_1, \ldots, x_n \in \mathcal{X}^*$ is $\mu_x(x^n) := |\{j : x_j = x, 1 \leq j \leq n\}|$, the number of times $x$ appears in $x^n$. These multiplicities induce an *empirical distribution* $p_\mu(x^n)$ that associates a probability $\mu_x(x^n)/n$ with each symbol $x \in \mathcal{X}$.

The *prevalence* of an integer $i \geq 0$ in $x^n$ is the number $\varphi_i(x^n)$ of symbols appearing $i$ times in $x^n$. For known $\mathcal{X}$, the value of $\varphi_0$ can be deduced from the remaining multiplicities, hence we define the *profile* of $x^n$ to be $\varphi(x^n) = (\varphi_1(x^n), \ldots, \varphi_n(x^n))$, the vector of all positive prevalences. For example, $\varphi(alfalfa) = (0, 2, 1, 0, 0, 0, 0)$. Note that the profile of $x^n$ also corresponds to the multiset of multiplicities of distinct symbols in $x^n$.

For a distribution $p \in \Delta_\mathcal{X}$, let

$$p(x^n) := \Pr_{X^n \sim p} (X^n = x^n)$$

be the probability of observing a sequence $x^n$ under i.i.d. sampling from $p$, and let

$$p(\varphi) := \sum_{y^n : \varphi(y^n) = \varphi} p(y^n)$$

be the probability of observing a profile $\varphi$. While the sequence maximum likelihood estimator maps a sequence to its empirical distribution, which maximizes the sequence probability $p(x^n)$, the *profile maximum likelihood (PML)* estimator [58] over a set $\mathcal{P} \subseteq \Delta_\mathcal{X}$ maps each profile $\varphi$ to a distribution

$$p_\varphi := \arg\max_p p(\varphi)$$

that maximizes the profile probability. Relaxing the optimization objective, for any $\beta \in (0, 1)$, a $\beta$-*approximate PML* estimator [4] maps each profile $\varphi$ to a distribution $p_\varphi^\beta$ such that $p_\varphi^\beta(\varphi) \geq \beta \cdot p_\varphi(\varphi)$.

Originating from the principle of maximum likelihood, PML was proved [2, 4, 6, 7, 25, 58] to possess a number of useful attributes, such as existence over finite discrete domains, majorization by empirical distributions, consistency for distribution estimation under both sorted and unsorted $\ell_1$ distances, and competitiveness to other profile-based estimators.

Let $\varepsilon$ be an error parameter and $f$ *be one of the four properties* in the introduction. Set $n := n_f(\varepsilon)$. Recently, Acharya et al. [4] showed that for some absolute constant $c' > 0$, if $c < c'$ and $\varepsilon \geq n^{-c}$, then a plug-in estimator for $f$, using an $\exp(-n^{1-\Theta(c)})$-approximate PML, is sample-optimal. Motivated by this result, Charikar et al. [20] constructed an explicit $\exp(-\mathcal{O}(n^{2/3} \log^3 n))$-approximate PML (APML) whose computation time is near-linear in $n$. Combined, these results provide a unified, sample-optimal, and near-linear-time computable plug-in estimator for the four properties.

## 2 New results and implications

### 2.1 New results

**Additive property estimation**

Recall that for any property $f$, the expression $n_f(\varepsilon)$ denotes the smallest sample size required by any estimator to achieve accuracy $\varepsilon$ with confidence $2/3$, for all distributions in $\Delta_{\mathcal{X}}$. Let $f$ be an additive symmetric property that is 1-Lipschitz on $(\Delta_{\mathcal{X}}, R)$. Let $\varepsilon > 0$ and $n \geq n_f(\varepsilon)$ be error and sampling parameters. For an absolute constant $c \in (10^{-2}, 10^{-1})$, if $\varepsilon \geq n^{-c}$,

**Theorem 1.** *The PML plug-in estimator, when given a sample of size $4n$ from any distribution $p \in \Delta_{\mathcal{X}}$, will estimate $f(p)$ up to an error of $(2 + o(1))\varepsilon$, with probability at least $1 - \exp\left(-4\sqrt{n}\right)$.*

For a different $c > 0$, Theorem 1 also holds for APML, which is near-linear-time computable [20].

**Rényi entropy estimation**

For $\mathcal{X}$ of finite size $k$ and any $p \in \Delta_{\mathcal{X}}$, it is well-known that $H_\alpha(p) \in [0, \log k]$. The following theorems characterize the performance of the PML plug-in estimator in estimating Rényi entropy.

For any distribution $p \in \Delta_{\mathcal{X}}$, error parameter $\varepsilon \in (0, 1)$, absolute constant $\lambda \in (0, 0.1)$, and sampling parameter $n$, draw a sample $X^n \sim p$ and denote its profile by $\varphi$. Then for sufficiently large $k$,

**Theorem 2.** *For $\alpha \in (3/4, 1)$, if $n = \Omega_\alpha(k^{1/\alpha}/(\varepsilon^{1/\alpha} \log k))$,*
$$\Pr\left(|H_\alpha(p_\varphi) - H_\alpha(p)| \geq \varepsilon\right) \leq \exp(-\sqrt{n}).$$

**Theorem 3.** *For non-integer $\alpha > 1$, if $n = \Omega_\alpha(k/(\varepsilon^{1/\alpha} \log k))$,*
$$\Pr\left(|H_\alpha(p_\varphi) - H_\alpha(p)| \geq \varepsilon\right) \leq \exp(-n^{1-\lambda}).$$

**Theorem 4.** *For integer $\alpha > 1$, if $n = \Omega_\alpha(k^{1-1/\alpha}(\varepsilon^2 \log(1/\varepsilon))^{-(1+\alpha)})$ and $H_\alpha(p) \leq (\log n)/4$,*
$$\Pr(|H_\alpha(p_\varphi) - H_\alpha(p)| \geq \varepsilon) \leq 1/3.$$

Replacing $3/4$ by $5/6$, Theorem 2 also holds for APML with a better probability bound $\exp(-n^{2/3})$. In addition, Theorem 3 holds for APML without any modifications.

**Sorted distribution estimation**

Let $c$ be the absolute constant defined just prior to Theorem 1. For any distribution $p \in \Delta_{\mathcal{X}}$, error parameter $\varepsilon \in (0, 1)$, and sampling parameter $n$, draw a sample $X^n \sim p$ and denote its profile by $\varphi$.

**Theorem 5.** *If $n = \Omega(n(\varepsilon)) = \Omega\left(k/(\varepsilon^2 \log k)\right)$ and $\varepsilon \geq n^{-c}$,*
$$\Pr(\ell_1^<(p_\varphi, p) \geq \varepsilon) \leq \exp(-\Omega(\sqrt{n})).$$

For a *different* $c > 0$, Theorem 5 also holds for APML with a better probability bound $\exp(-n^{2/3})$.

**Identity testing**

The recent works of Diakonikolas and Kane [26] and Goldreich [31] provided a procedure reducing identity testing to uniformity testing, while modifying the desired accuracy and alphabet size by only absolute constant factors. Hence below we consider uniformity testing.

The uniformity tester $T_{\text{PML}}$ shown in Figure 1 is purely based on PML and satisfies

**Theorem 6.** *If $\varepsilon = \tilde{\Omega}(k^{-1/4})$ and $n = \tilde{\Omega}(\sqrt{k}/\varepsilon^2)$, then the tester $T_{PML}(X^n)$ will be correct with probability at least $1 - k^{-2}$. The tester also distinguishes between $p = p_u$ and $\|p - p_u\|_2 \geq \varepsilon/\sqrt{k}$.*

The $\tilde{\Omega}(\cdot)$ notation only hides logarithmic factors of $k$. The tester $T_{\text{PML}}$ is near-optimal as for uniform distribution $p_u$, the results in [28] yield an $\Omega(\sqrt{k \log k}/\varepsilon^2)$ lower bound on $n_{p_u}(\varepsilon, k^{-2})$.

For space considerations, we postpone proofs and additional results to the supplementary material. The rest of the paper is organized as follows. Section 2.2 presents several immediate implications of the above theorems. Section 3 and Section 4 illustrate PML's theoretical and practical advantages by comparing it to existing methods for a variety of learning tasks. Section 5 concludes the paper and outlines multiple promising future directions.

> **Input:** parameters $k, \varepsilon$, and a sample $X^n \sim p$ with profile $\varphi$.
> **if** $\max_x \mu_x(X^n) \geq 3 \max\{1, n/k\} \log k$ **then return** 1;
> **elif** $\|p_\varphi - p_u\|_2 \geq 3\varepsilon/(4\sqrt{k})$ **then return** 1;
> **else return** 0.

Figure 1: Uniformity tester $T_{\mathrm{PML}}$

## 2.2 Implications

Several immediate implications are in order.

We say that a plug-in estimator is *universally sample-optimal* for estimating symmetric properties if there exist absolute positive constants $c_1, c_2$ and $c_3$, such that for any 1-Lipschitz property on $(\Delta_\mathcal{X}, R)$, with probability $\geq 9/10$, the plug-in estimator uses just $c_1$ times the sample size $n$ required by the minimax estimator to achieve $c_2$ times its error, whenever this error is at least $n^{-c_3}$.

Note that the "1-Lipschitz property" class can be replaced by other general property classes, but not by those containing only a few specific properties, since "universal" means "applicable to all cases".

**Theorem 1** makes PML the *first* plug-in estimator that is *universally sample-optimal* for a broad class of distribution properties. In particular, Theorem 1 also covers the four properties considered in [4]. To see this, as mentioned in the introduction, $\tilde{C}_m$, $H$, and $D$ are 1-Lipschitz on $(\Delta_\mathcal{X}, R)$; as for $\tilde{S}$, the following result [4] relates it to $\tilde{C}_m$ for distributions in $\Delta_{\geq 1/k}$, and proves PML's optimality.

**Lemma 1.** *For any $\varepsilon > 0$, $m = k \log(1/\varepsilon)$, and $p \in \Delta_{\geq 1/k}$,*

$$|\tilde{S}(p) - \tilde{C}_m(p) \log (1/\varepsilon)| \leq \varepsilon.$$

The theorem also applies to many other properties. As an example [70], given an integer $s > 0$, let $f_s(x) := \min\{x, |x - 1/s|\}$. Then to within a factor of two, $f_s(p) := \sum_x f_s(p_x)$ approximates the $\ell_1$ distance between any distribution $p$ and the closest uniform distribution in $\Delta_\mathcal{X}$ of support size $s$.

In Section 3.2 we compare Theorem 1 with existing results and present more of its implications.

**Theorem 2 and 3** imply that for all non-integer $\alpha > 3/4$ (resp. $\alpha > 5/6$), the PML (resp. APML) plug-in estimator achieves a sample complexity better than the best currently known [5]. This makes both the PML and APML plug-in estimators the state-of-the-art algorithms for estimating non-integer order Rényi entropy. See Section 3.3 for an introduction of known results, and see Section 3.4 for a detailed comparison between existing methods and ours.

**Theorem 4** shows that for all integer $\alpha > 1$, the sample complexity of the PML plug-in estimator has optimal $k^{1-1/\alpha}$ dependence [5, 55] on the alphabet size.

**Theorem 5** makes APML the first distribution estimator under sorted $\ell_1$ distance that is both near-linear-time computable and sample-optimal for a range of desired accuracy $\varepsilon$ beyond inverse poly-logarithmic of $n$. In comparison, existing algorithms [2, 38, 72] either run in polynomial time in the sample sizes, or are only known to achieve optimal sample complexity for $\varepsilon = \Omega(1/\sqrt{\log n})$, which is essentially different from the applicable range of $\varepsilon \geq n^{-\Theta(1)}$ in Theorem 5. We provide a more detailed comparison in Section 3.6.

**Theorem 6** provides the first PML-based uniformity tester with near-optimal sample complexity. As stated, the tester also distinguishes between $p = p_u$ and $\|p - p_u\|_2 \geq \varepsilon/\sqrt{k}$. This is a stronger guarantee since by the Cauchy-Schwarz inequality, $\|p - p_u\|_1 \geq \varepsilon$ implies $\|p - p_u\|_2 \geq \varepsilon/\sqrt{k}$.

Note that several other uniformity testers in the literature (see Section 3.7) also provide the same $\ell_2$ testing guarantee, since all of them are essentially counting sample collisions, i.e., the number of location pairs such that the sample points at those locations are equal.

# 3 Related work and comparisons

## 3.1 Additive property estimation

The study of additive property estimation dates back at least half a century [16, 34, 35] and has steadily grown over the years. For any additive symmetric property $f$ and sequence $x^n$, the simplest and most widely-used approach uses the *empirical (plug-in) estimator* $\hat{f}^E(x^n) := f(p_\mu(x^n))$ that evaluates $f$ at the empirical distribution. While the empirical estimator performs well in the large-sample regime, modern data science applications often concern high-dimensional data, for which more involved methods have yielded property estimators that are more sample-efficient. For example, for relatively large $k$ and for $f$ being $\tilde{S}, \tilde{C}_m, H$, or $D$, recent research [45, 59, 69, 70, 75, 76] showed that the empirical estimator is optimal up to logarithmic factors, namely $n_f(\mathcal{P}, \varepsilon) = \Theta_\varepsilon(n_f(\hat{f}^E, \mathcal{P}, \varepsilon)/\log n_f(\hat{f}^E, \mathcal{P}, \varepsilon))$, where $\mathcal{P}$ is $\Delta_{\geq 1/k}$ for $\tilde{S}$, and is $\Delta_\mathcal{X}$ for the other properties.

Below we classify the methods for deriving the corresponding sample-optimal estimators into two categories: plug-in and approximation, and provide a high-level description. For simplicity of illustration, we assume that $\varepsilon \in (0, 1]$.

The *plug-in* approach essentially estimates the unknown distribution multiset, which suffices for computing any symmetric properties. Besides the empirical and PML estimators, Efron and Thisted [29] proposed a linear-programming approach that finds a multiset estimate consistent with the sample's profile. This approach was then adapted and analyzed by Valiant and Valiant [69, 72], yielding plug-in estimators that achieve near-optimal sample complexities for $H$ and $\tilde{S}$, and optimal sample complexity for $D$, when $\varepsilon$ is relatively large.

The *approximation* approach modifies non-smooth segments of the probability function to correct the bias of empirical estimators. A popular modification is to replace those non-smooth segments by their low-degree polynomial approximations and then estimate the modified function. For several properties including the above four and *power sum* $P_\alpha(p) := \sum_x p_x^\alpha$, where $\alpha$ is a given parameter, this approach yields property-dependent estimators [45, 59, 75, 76] that are sample-optimal for all $\varepsilon$.

More recently, Acharya et al. [4] proved the aforementioned results on PML estimator and made it the first unified, sample-optimal plug-in estimator for $\tilde{S}, \tilde{C}_m, H$ and $D$ and relatively large $\varepsilon$. Following these advances, Han et al. [38] refined the linear-programming approach and designed a plug-in estimator that implicitly performs polynomial approximation and is sample-optimal for $H, \tilde{S}$, and $P_\alpha$ with $\alpha < 1$, when $\varepsilon$ is relatively large.

## 3.2 Comparison I: Theorem 1 and related property-estimation work

In terms of the estimator's theoretical guarantee, Theorem 1 is essentially the same as Valiant and Valiant [70]. However, for each property, $k$, and $n$, [70] solves a different linear program and constructs a new estimator, which takes polynomial time. On the other hand, both the PML estimator and its near-linear-time computable variant, once computed, can be used to accurately estimate exponentially many properties that are 1-Lipschitz on $(\Delta_\mathcal{X}, R)$. A similar comparison holds between the PML method and the approximation approach, while the latter is provably sample-optimal for only a few properties. In addition, Theorem 1 shows that the PML estimator often achieves the optimal sample complexity up to a small constant factor, which is a desired estimator attribute shared by some, but not all approximation-based estimators [45, 59, 75, 76].

In term of the method and proof technique, Theorem 1 is closest to Acharya et al. [4]. On the other hand, [4] establishes the optimality of PML for only four properties, while our result covers a much broader property class. In addition, both the above mentioned "small constant factor" attribute, and the confidence boost from $2/3$ to $1 - \exp(-4\sqrt{n})$ are unique contributions of this work. The PML plug-in approach is also close in flavor to the plug-in estimators in Valiant and Valiant [69, 72] and their refinement in Han et al. [38]. On the other hand, as pointed out previously, these plug-in estimators are provably sample-optimal for only a few properties. More specifically, for estimating $H$, $\tilde{S}$, and $\tilde{C}_m$, the plug-in estimators in [69, 72] achieve sub-optimal sample complexities with regard to the desired accuracy $\varepsilon$; and the estimation guarantee in [38] is provided in terms of the approximation errors of $\tilde{\mathcal{O}}(\sqrt{n})$ polynomials that are not directly related to the optimal sample complexities.

### 3.3 Rényi entropy estimation

Motivated by the wide applications of Rényi entropy, heuristic estimators were proposed and studied in the physics literature following [36], and asymptotically consistent estimators were presented and analyzed in the statistical learning literature [46, 78]. For the special case of 1-Rényi (or Shannon) entropy, the works of [69, 70] determined the sample complexity to be $n_f(\varepsilon) = \Theta(k/(\varepsilon \log k))$.

For general $\alpha$-Rényi entropy, the best-known results in Acharya et al. [5] state that for integer and non-integer $\alpha$ values, the corresponding sample complexities $n_f(\varepsilon, \delta)$ are $\mathcal{O}_\alpha(k^{1-1/\alpha} \log(1/\delta)/\varepsilon^2)$ and $\mathcal{O}_\alpha(k^{\min\{1/\alpha,1\}} \log(1/\delta)/(\varepsilon^{1/\alpha} \log k))$, respectively. The upper bounds for integer $\alpha$ are achieved by an estimator that corrects the bias of the empirical plug-in estimator. To achieve the upper bounds for non-integer $\alpha$ values, one needs to compute some best polynomial approximation of $z^\alpha$, whose degree and domain both depend on $n$, and construct a more involved estimator using the approximation approach [45, 75] mentioned in Section 3.1.

### 3.4 Comparison II: Theorem 2 to 4 and related Rényi-entropy-estimation work

Our result shows that a single PML estimate suffices to estimate the Rényi entropy of different orders $\alpha$. Such adaptiveness to the order parameter is a significant advantage of PML over existing methods. For example, by Theorem 3 and the union bound, one can use a *single* APML or PML to accurately approximate exponentially many non-integer order Rényi entropy values, yet still maintains an overall confidence of $1 - \exp(-k^{0.9})$. By comparison, the estimation heuristic in [5] requires different polynomial-based estimators for different $\alpha$ values. In particular, to construct each estimator, one needs to compute some best polynomial approximation of $z^\alpha$, which is not known to admit a closed-form formula for $\alpha \notin \mathbb{Z}$. Furthermore, even for a single $\alpha$ and with a sample size $\sqrt{k}$ times larger, such estimator is not known to achieve the same level of confidence as PML or APML.

As for the theoretical guarantees, the sample-complexity upper bounds in both Theorem 2 and 3 are better than those mentioned in the previous section. More specifically, for any $\alpha \in (3/4, 1)$ and $\delta \geq \exp(-k^{-0.5})$, Theorem 2 shows that $n_f(\varepsilon, \delta) = \mathcal{O}_\alpha(k^{1/\alpha}/(\varepsilon^{1/\alpha} \log k))$. Analogously, for any non-integer $\alpha > 1$ and $\delta \geq \exp(-k^{-0.9})$, Theorem 3 shows that $n_f(\varepsilon, \delta) = \mathcal{O}_\alpha(k/(\varepsilon^{1/\alpha} \log k))$. Both bounds are better than the best currently known by a $\log(1/\delta)$ factor.

### 3.5 (Sorted) distribution estimation

Estimating large-alphabet distributions from their samples is a fundamental statistical learning tenet. Over the past few decades, distribution estimation has found numerous applications, ranging from natural language modeling [21] to biological research [8], and has been studied extensively. Under the classical $\ell_1$ and KL losses, existing research [13, 47] showed that the corresponding sample complexities $n(\varepsilon)$ are $\Theta(k/\varepsilon^2)$ and $\Theta(k/\varepsilon)$, respectively. Several recent works have investigated the analogous formulation under sorted $\ell_1$ distance, and revealed a lower sample complexity of $n(\varepsilon) = \Theta(k/(\varepsilon^2 \log k))$. Specifically, under certain conditions, Valiant and Valiant [72], Han et al. [38] derived sample-optimal estimators using linear programming, and Acharya et al. [2], Das. [25] showed that PML achieves a sub-optimal $\mathcal{O}(k/(\varepsilon^{2.1} \log k))$ sample complexity for relatively large $\varepsilon$.

### 3.6 Comparison III: Theorem 5 and related distribution-estimation work

We compare our results with existing ones from three different perspectives.

**Applicable parameter ranges:** As shown by [38], for $\varepsilon \ll n^{-1/3}$, the simple empirical estimator is already sample-optimal. Hence we consider the parammeter range $\varepsilon = \Omega(n^{-1/3})$. For the results in [2, 25] and [72] to hold, we would need $\varepsilon$ to be at least $\Omega(1/\sqrt{\log n})$. On the other hand, Theorem 5 shows that PML and APML are sample-optimal for $\varepsilon$ larger than $n^{-\Theta(1)}$. Here, the gap is exponentially large. The result in [38] applies to the whole range $\varepsilon = \Omega(n^{-1/3})$, which is larger than the applicable range of our results.

**Time complexity:** Both the APML and the estimator in [72] are near-linear-time computable in the sample sizes, while the estimator in [38] would require polynomial time to be computed.

**Statistical confidence:** The PML and APML achieve the desired accuracy with an error probability at most $\exp(-\Omega(\sqrt{n}))$. On the contrary, the estimator in [38] is known to achieve an error probability

that decreases only as $\mathcal{O}(n^{-3})$. The gap is again exponentially large. The estimator in [72] admits a better error probability bound of $\exp(-n^{0.02})$, which is still far from ours.

## 3.7 Identity testing

Initiated by the work of [33], identity testing is arguably one of the most important and widely-studied problem in distribution property testing. Over the past two decades, a sequence of works [3, 11, 26–28, 33, 61, 68] have addressed the sample complexity of this problem and proposed testers with a variety of guarantees. In particular, applying a coincidence-based tester, Paninski [61] determined the sample complexity of uniformity testing up to constant factors; utilizing a variant of the Pearson chi-squared statistic, Valiant and Valiant [68] resolved the general identity testing problem. For an overview of related results, we refer interested readers to [15] and [32]. The contribution of this work is mainly showing that PML, is a unified sample-optimal approach for several related problems, and as shown in Theorem 6, also provides a near-optimal tester for this important testing problem.

## 4 Experiments and distribution estimation under $\ell_1$ distance

A number of different approaches have been taken to computing the PML and its approximations. Among the existing works, Acharya et al. [1] considered exact algebraic computation, Orlitsky et al. [57, 58] designed an EM algorithm with MCMC acceleration, Vontobel [73, 74] proposed a Bethe approximation heuristic, Anevski et al. [7] introduced a sieved PML estimator and a stochastic approximation of the associated EM algorithm, and Pavlichin et al. [62] derived a dynamic programming approach. Notably and recently, for a sample size $n$, Charikar et al. [20] constructed an explicit $\exp(-\mathcal{O}(n^{2/3}\log^3 n))$-approximate PML whose computation time is near-linear in $n$.

In Section 4 of the supplementary material, we introduce a variant of the MCMC-EM algorithm in [60] and demonstrate the exceptional efficacy of PML on a variety of learning tasks through experiments. In particular, we derive a new distribution estimator for (unsorted) $\ell_1$ distance by combining the proposed PML computation algorithm with the denoising procedure in [71] and a novel missing mass estimator. As shown below, the proposed distribution estimator has the state-of-the-art performance.

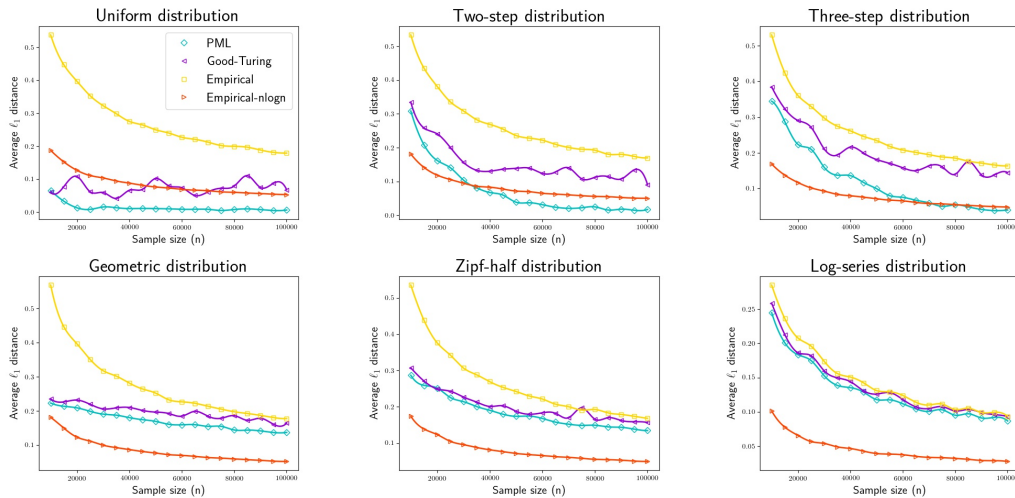

Figure 2: Distribution estimation under $\ell_1$ distance

In Figures 2, samples are generated according to six distributions of the same support size $k = 5,000$. Details about these distributions can be found in Section 4.2 of the supplementary material. The sample size $n$ (horizontal axis) ranges from 10,000 to 100,000, and the vertical axis reflects the (unsorted) $\ell_1$ distance between the true distribution and the estimates, averaged over 30 independent trials. We compare our estimator with three different ones: the improved Good-Turing estimator in [56, 41], which is provably instance-by-instance near-optimal [56], the empirical estimator, serving as a baseline, and the empirical estimator with a larger $n \log n$ sample size. Note that $\log n$ is

roughly 11. As shown in [56], the improved Good-Turing estimator substantially outperformed other estimators such as the Laplace (add-1) estimator, the Braess-Sauer estimator [13], and the Krichevsky-Trofimov estimator [48]. Hence we do not include those estimators here. The following plots showed that our proposed estimator further outperformed the improved Good-Turing estimator in all the experiments.

## 5 Conclusion and future directions

We studied three fundamental problems in statistical learning: distribution estimation, property estimation, and property testing. We established the profile maximum likelihood (PML) as the first universally sample-optimal approach for several important learning tasks: distribution estimation under the sorted $\ell_1$ distance, additive property estimation, Rényi entropy estimation, and identity testing. Several future directions are promising. We believe that neither the factor of $4$ in the sample size in Theorem 1, nor the lower bounds on $\varepsilon$ in Theorem 1, 5, and 6 are necessary. In other words, the PML approach is universally sample-optimal for these tasks in all ranges of parameters. It is also of interest to extend the PML's optimality to estimating symmetric properties not covered by Theorem 1 to 4, such as *generalized distance to uniformity* [9, 39], the $\ell_1$ distance between the unknown distribution and the closest uniform distribution over an arbitrary subset of $\mathcal{X}$.

Another important direction is *competitive (or instance-optimal) property estimation*. It should be noted that all the referenced works including this paper are of the worst-case nature, namely, designing estimators with near-optimal worst-case performances. On the contrary, practical and natural distributions often possess simple structures, and are rarely the worst possible. To address this discrepancy, the recent work [40, 43] took a competitive approach and constructed estimators whose performances are adaptive to the simplicity of the underlying distributions. Specifically, for any property in a broad class and *every* distribution in $\Delta_{\mathcal{X}}$, the expected error of the proposed estimator with a sample of size $n/\log n$ is at most that of the empirical estimator with a sample of size $n$, pluses a distribution-free vanishing function of $n$. These results not only cover $\tilde{S}, \tilde{C}_m, H$, and $D$, for which the $\log n$-factor is optimal up to constants, but also apply to any *non-symmetric additive* property $\sum_x f_x(p_x)$ where $f_x$ is 1-Lipschitz for all $x \in \mathcal{X}$, such as the $\ell_1$-distance to a given distribution. It would be of interest to study the optimality of the PML approach under this formulation as well. Readers interested in estimating non-symmetric properties may also find the paper [42] helpful.

### Acknowledgments

We are grateful to the National Science Foundation (NSF) for supporting this work through grants CIF-1564355 and CIF-1619448.

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
