[Supplementary Material]

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

## 4.1   MCMC-EM algorithm variant

To approximate PML, the work [65] proposed an MCMC-EM algorithm, where MCMC and EM stand for Markov chain Monte Carlo and expectation maximization, respectively. A sketch of the original MCMC-EM algorithm can be found in [65], and a detailed description is available in Chapter 6 of [69]. The EM part uses a simple iteration procedure to update the distribution estimates. One can show [69] that it is equivalent to the conventional *generalized gradient ascent method*. The MCMC part exploits local properties of the update process and accelerates the EM computation. Below we present a variant of this algorithm that often runs faster and is more accurate.

**Step 1:**   We separate the large and small multiplicities. Define a threshold parameter $\tau := 1.5 \log^2 n$ and suppress $X^n$ in $p_\mu(X^n)$ for simplicity. For symbols $x$ with $\mu_x(X^n) \geq \tau$, estimate their probabilities by $p_\mu(x) = \mu_x(X^n)/n$ and remove them from the sample. Denote the collection of removed symbols by $R$ and the remaining sample sequence by $X^r$. In the subsequent steps, we apply the EM-MCMC algorithm to $X^r$.

The idea is simple: By the Chernoff-type bound for binomial random variables, with high probability, the empirical frequency $\mu_x(X^n)/n$ of a large-multiplicity symbol $x$ is very close to its mean value $p(x)$. Hence for large-multiplicity symbols we can simply use the empirical estimates and focus on estimating the probabilities of small-multiplicity symbols. This is similar to initializing the EM algorithm by the empirical distribution and fixing the large probability estimates through the iterations. However, the approach described here is more efficient.

**Step 2:**   We determine a proper alphabet size for the output distribution of the EM algorithm. If the true value $k$ is provided, then we simply use $k - |R|$. Otherwise, we apply the following support size estimator [5] to $X^r$:

$$\hat{S}(X^r) := \sum_{j \geq 1} (1 - (-(t-1))^j \Pr(L \geq j)) \cdot \varphi_j(X^r),$$

where $t = \log r$ and $L$ is an independent binomial random variable with support size $\lceil \frac{1}{2} \log_2(\frac{rt^2}{t-1}) \rceil$ and success probability $(t+1)^{-1}$. For any $\varepsilon$ larger than an absolute constant, estimator $\hat{S}$ achieves the optimal sample complexity $n_f(\Delta_{\geq 1/k}, \varepsilon)$ in estimating support size, up to constant factors [5].

**Step 3:**   Apply the MCMC-EM algorithm in [65, 69] to $\varphi(X^r)$ with the output alphabet size determined in the previous step, and denote the resulting distribution estimate by $p_r$. (In the experiments, we perform the EM iteration for 30 times.) Intuitively, this estimate corresponds to the conditional distribution given that the next observation is a symbol with small probability.

**Step 4:**   Let $T_\mu := \sum_{x \in R} p_\mu(x)$ be the total probability of the large-multiplicity symbols. Treat $p_r$ as a vector and let $p_r' := (1 - T_r) \cdot p_r$. For every symbol $x \in R$, append $p_\mu(x)$ to $p_r'$, and return the resulting vector. Note that this vector corresponds to a valid discrete distribution.

**Algorithm code**

The implementation of our algorithm is also available at https://github.com/ucsdyi/PML.

For computational efficiency, the program code for the original MCMC-EM algorithm in [65, 69] is written in C++, with a file name "MCMCEM.cpp". The program code for other functions is written in Python3. Note that to execute the program, one should have a 64-bit Windows/Linux system with Python3 installed (64-bit version). In addition, we also use functions provided by "NumPy" and "SciPy", while the latter is not crucial and can be removed by modifying the code slightly.

Our implementation also makes use of "ctypes", a *built-in* foreign language library for Python that allows us to call C++ functions directly. Note that before calling C++ functions in Python, we need to compile the corresponding C++ source files into DLLs or shared libraries. We have compiled and included two such files, one is "MCMCEM.so", the other is "MCMCEM.dll".

Functions in "MCMCEM.cpp" can be used separately. To compute a PML estimate, simply call the function "int PML(int MAXSZ=10000, int maximum_EM=20, int EM_n=100)", where the first parameter specifies an upper bound on the support size of the output distribution, the second provides the maximum number of EM iteration, and the last corresponds to the sample size $n$. This function takes as input a local file called "proFile", which contains the profile vector $\varphi(X^n)$ in the format of "1 4 7 10 ...". Specifically, the file "proFile" consists of only space-separated non-negative integers, and the $i$-th integer represents the value of $\varphi_i(X^n)$. The output is a vector of length at most MAXSZ, and is stored in another local file called "PMLFile". Each line of the file "PMLFile" contains a non-negative real number, corresponding to a probability estimate.

To perform experiments and save the plots to the directory containing the code, simply execute the file "Main.py". To avoid further complication, the code compares our estimator with only three other estimators: empirical, empirical with a larger $n \log n$ sample size, and improved Good-Turing [64] (for distribution estimation under unsorted $\ell_1$ distance). The implementation covers all the distributions described in the next section. One can test any of these distributions by including it in "D_List" of the "main()" function. The implementation also covers a variety of learning tasks, such as distribution estimation under sorted and unsorted $\ell_1$ distances, and property estimation for Shannon entropy, $\alpha$-Rényi entropy, support coverage, and support size.

Finally, functions related to distribution and sample generation are available in file "Samples.py". Others including the property computation functions, the sorted and unsorted $\ell_1$ distance functions, and the previously-described support size estimator, are contained in file "Functions.py".

## 4.2 Experiment distributions

In the following experiments, samples are generated according to six distributions with the same support size $k = 5,000$.

Three of them have finite support by definition: uniform distribution, two-step distribution with half the symbols having probability $2/(5k)$ and the other half have probability $8/(5k)$, and a three-step distribution with one third the symbols having probability $3/(13k)$, another third having probability $9/(13k)$, and the remaining having probability $27/(13k)$.

The other three distributions are over $\{i \in \mathbb{Z} : i \geq 1\}$, and are truncated at $i = 5,000$ and re-normalized: geometric distribution with parameter $g = 1/k$ satisfying $p_i \propto (1 - g)^i$, Zipf distribution with parameter $1/2$ satisfying $p_i \propto i^{-1/2}$, and log-series distribution with parameter $\gamma = 2/k$ satisfying $p_i \propto (1 - \gamma)^i/i$.

## 4.3 Experiment results and details

As shown below, the proposed PML approximation algorithm has exceptional performance.

**Distribution estimation under $\ell_1$ distance**

We derive a new distribution estimator under the (unsorted) $\ell_1$ distance by combining the proposed PML computation algorithm with the denoising procedure in [81] and a missing mass estimator [64].

First we describe this distribution estimator, which takes a sample $X^n$ from some unknown distribution $p$. An optional input is $\mathcal{X}$, the underlying alphabet.

**Step 1:**  Apply the PML computation algorithm described in Section 4.1 to $X^n$, and denote the returned vector, consisting of non-negative real numbers that sum to 1, by $V$.

**Step 2:**  Employ the following variant of the denoising procedure in [81]. Arbitrarily remove a total probability mass of $\log^{-2} n$ from entries of the vector $V$ without making any entry negative. Then for each $j \leq \log^2 n$, augment the vector by $n/(j \log^4 n)$ entries of probability $j/n$. For every multiplicity $\mu \geq 1$ appearing in the sample, assign to all symbols appearing $\mu$ times the following probability value. If $\mu \geq \log^2 n$, simply assign to each of these symbols the empirical estimate $\mu/n$; otherwise, temporally associate a weight of $\text{bin}(n, v, \mu) := \binom{n}{\mu}(1-v)^{n-\mu}v^\mu$ with each entry $v$ in $V$, and assign to each of these symbols the current weighted median of $V$.

**Step 3:**  If $\mathcal{X}$ is available, we can estimate the total probability mass $M(X^n) := \sum_{x \in \mathcal{X}} \mathbb{1}_{x \notin X^n}$ of the unseen symbols (a.k.a., the *missing mass*) by the following estimator:

$$\hat{M}(X^n) := \frac{\varphi_1(X^n)}{\sum_j (j\varphi_j(X^n)\mathbb{1}_{j>\varphi_{j+1}} + (j+1)\varphi_{j+1}(X^n)\mathbb{1}_{j\leq\varphi_{j+1}})}.$$

We equally distribute this probability mass estimate among symbols that do not appear in the sample.

As shown below, the proposed distribution estimator achieves the state-of-the-art performance.

In Figures 2, the horizontal axis reflects the sample size $n$, ranging from 10,000 to 100,000, and the vertical axis reflects the (unsorted) $\ell_1$ distance between the true distribution and the estimates, averaged over 30 independent trials. We compare our estimator with three others: the improved Good-Turing estimator [64, 46], the empirical estimator, serving as a baseline, and the empirical estimator with a larger $n \log n$ sample size. Note that $\log n$ is roughly 11. As shown in [64], the improved Good-Turing estimator is provably instance-by-instance near-optimal and substantially outperforms other estimators such as the Laplace (add-1) estimator, the Braess-Sauer estimator [14], and the Krichevsky-Trofimov estimator [55]. Hence we do not include those estimators below.

As the following plots show, our proposed estimator outperformed the improved Good-Turing estimator in all experiments.

Figure 2: Distribution estimation under $\ell_1$ distance

**Distribution estimation under sorted $\ell_1$ distance**

In Figure 3, the sample size $n$ ranges from 2,000 to 20,000, and the vertical axis reflects the sorted $\ell_1$ distance between the true distribution and the estimates, averaged over 30 independent trials. We compare our estimator with that proposed by Valiant and Valiant [82] that utilizes linear programming, with the empirical estimator, and with the empirical estimator with a larger $n \log n$ sample size.

We do not include the estimator in [43] since there is no implementation available, and as pointed out by the recent work of [84] (page 7), the approach in [43] "is quite unwieldy. It involves significant parameter tuning and special treatment for the edge cases." and "Some techniques ... are quite crude and likely lose large constant factors both in theory and in practice."

As shown in Figure 3, with the exception of uniform distribution, where the estimator in Valiant and Valiant [82] (VV-LP) is the best and PML is the closest second, the PML estimator outperforms VV-LP for all other tested distributions. As the underlying distribution becomes more skewed, the improvement of PML over VV-LP grows. For the log-series distribution, the performance of VV-LP is even worse than the empirical estimator.

Additionally, the plots also demonstrate that PML has a more stable performance than VV-LP.

Figure 3: Distribution estimation under sorted $\ell_1$ distance

**Shannon entropy estimation under absolute error**

In Figure 4, the sample size $n$ ranges from $1,000$ to $1,000,000$, and the vertical axis reflects the absolute difference between the true entropy values and the estimates, averaged over 30 independent trials. We compare our estimator with two state-of-the-art estimators, *WY* in [87], and *JVHW* in [51], as well as the empirical estimator, and the empirical estimator with a larger $n \log n$ sample size. Additional entropy estimators such as the Miller-Mallow estimator [18], the best upper bound (BUB) estimator [70], and the Valiant-Valiant estimator [82] were compared in [87, 51] and found to perform similarly to or worse than the two estimators that we compared with, therefore we do not include them here. Also, considering [82], page 50 in [91] notes that "the performance of linear programming estimator starts to deteriorate when the sample size is very large."

Note that the alphabet size $k$ is a crucial input to WY, but is not required by either JVHW or our PML algorithm. In the experiments, we provide WY with the true value of $k = 5,000$.

As shown in the plots, our estimator performs as well as these state-of-the-art estimators.

Figure 4: Shannon entropy estimation under absolute error

**$\alpha$-Rényi entropy estimation under absolute error**

For a distribution $p \in \Delta_{\mathcal{X}}$, recall that the $\alpha$-power sum of $p$ is $P_{\alpha}(p) = \sum_x p(x)^{\alpha}$, implying $H_{\alpha}(p) = (1 - \alpha)^{-1} \log(P_{\alpha}(p))$. To establish the sample-complexity upper bounds mentioned in Section 3.3 for non-integer $\alpha$ values, Acharya et al. [6] first estimate the $P_{\alpha}(p)$ using the $\alpha$-power-sum estimator proposed in [51], and then substitute the estimate into the previous equation. The authors of [51] have implemented this two-step Rényi entropy estimation algorithm. In the experiments, we take a sample of size $n$, ranging from 10,000 to 100,000, and compare our estimator with this implementation, referred to as *JVHW*, the empirical estimator, and the empirical estimator with a larger $n \log n$ sample size. Note that $\log n$ ranges from 9.2 to 11.5. According to the results in [6], the sample complexities for estimating $\alpha$-Rényi entropy are quite different for $\alpha < 1$ and $\alpha > 1$, hence we consider two cases: $\alpha = 0.5$ and $\alpha = 1.5$.

As shown in Figure 5 and 6, our estimator clearly outperformed the one proposed by [6, 51].

We further note that for small sample sizes and several distributions, the estimator in [6, 51] performs significantly worse than ours. Also, for large sample sizes, the estimators in [6, 51] degenerates to the simple empirical plug-in estimator. In comparison, our proposed estimator tracks the performance of the empirical estimator with a larger $n \log n$ sample size for nearly all the tested distributions.

Figure 5: 0.5-Rényi entropy estimation under absolute error

Figure 6: 1.5-Rényi entropy estimation under absolute error

## 5 Lipschitz-property estimation

### 5.1 Proof outline of Theorem 1

The proof proceeds as follows. First, fixing $n$, $\mathcal{X}$, and a symmetric additive property $f$ that is 1-Lipschitz on $(\Delta_{\mathcal{X}}, R)$, we consider a related linear program defined in [83], and lower bound the worst-case error of any estimators using the linear program's objective value, say $v$. Second, following the construction in [83], we find an explicit estimator $\hat{f}^{\star}$ that is *linear*, i.e., can be expressed as a linear combination of $\varphi_i$'s, and show optimality by upper bounding its worst-case error in terms of $v$. Third, we study the concentration of a general linear estimator, and through the McDiarmid's inequality [60], relate the tail probability of its estimate to the estimator's sensitivity to the input changes. Fourth, we bound the sensitivity of $\hat{f}^{\star}$ by the maximum difference between its consecutive coefficients, and further bound this difference by a function of $n$, showing that the estimate induced by $\hat{f}^{\star}$ highly concentrates around its expectation. Finally, we invoke the result in [5] that the PML-plug-in estimator is competitive to all profile-based estimators whose estimates are highly concentrated, concluding that PML shares the optimality of $\hat{f}^{\star}$, thereby establishing Theorem 1.

### 5.2 Technical details

Let $f$ be a symmetric additive property that is 1-Lipschitz on $(\Delta_{\mathcal{X}}, R)$. Without loss of generality, we assume that $f(p) = 0$ if $p(x) = 1$ for some $x \in \mathcal{X}$.

**Lower bound**    First, fixing $n$, $\mathcal{X}$, and $f$, we lower bound the worst-case error of any estimators.

Let $u \in (0, 1/2)$ be a small absolute constant. If there is an estimator $\hat{f}$ that, when given a length-$n$ sample from any distribution $p \in \Delta_{\mathcal{X}}$, will estimate $f(p)$ up to an error of $\varepsilon$ with probability at least $1/2 + u$. Then for any two distributions $p_1, p_2 \in \Delta_{\mathcal{X}}$ satisfying $|f(p_1) - f(p_2)| > \varepsilon$, we can use $\hat{f}$ to distinguish $X^n \sim p_1$ from $X^n \sim p_2$, and will be correct with probability at least $1/2 + u$.

On the other hand, for any parameter $c_1 \in (1/100, 1/25]$ and $c_2 = 1/2 + 6c_1$, consider the corresponding linear program defined in Linear Program 6.7 in [83], and denote by $v$ the objective value of any of its solutions. Then, Proposition 6.8 in [83] implies that we can find two distributions $p_1, p_2 \in \Delta_{\mathcal{X}}$ such that $|f(p_1) - f(p_2)| > v \cdot (1 - o(1)) - O(n^{-c_1} \log n)$, and no algorithm can use $\mathrm{Poi}(n)$ sample points to distinguish these two distributions with probability at least $1/2 + u$.

The previous reasoning yields that $v < (1 + o(1))\varepsilon + O(n^{-c_1} \log n)$. By construction, $v$ is a function of $\mathcal{X}, n$, and $f$, and essentially serves as a lower bound for $\varepsilon$.

**Upper Bound**   Second, fixing $n$, $\mathcal{X}$, and $f$, we construct an explicit estimator based on the previously mentioned linear program, and show optimality by upper bounding its worst-case error in terms of $v$, the linear program's objective value.

A property estimator $\hat{f}$ is *linear* if there exist real coefficients $\{\ell_i\}_{i \geq 1}$ such that the identity $\hat{f}(x^n) = \sum_{i \geq 1} \ell_i \cdot \varphi_i(x^n)$ holds for all $x^n$. The following lemma (Proposition 6.10 in [83]) bounds the worst-case error of a linear estimator when its coefficients satisfy certain conditions.

**Lemma 2.** *Given any positive integer $m$, and real coefficients $\{\beta_i\}_{i \geq 0}$, define $\varepsilon(y) := f(y)/y - e^{-my} \sum_{i \geq 0} \beta_i \cdot (my)^i/i!$. Let $\beta_i^\star := \beta_{i-1} \cdot i/m, \forall i \geq 1$, and $\beta_0^\star := 0$. If for some $a', b', c' > 0$,*

1. $|\varepsilon(y)| \leq a' + b'/y$,

2. $|\beta_j^\star - \beta_\ell^\star| \leq c' \sqrt{j/m}$ *for any $j$ and $\ell$ such that $|j - \ell| \leq \sqrt{j} \log m$,*

*then given a sample $X^m$ from any $p \in D_\mathcal{X}$, the estimator defined by $\sum_{i \geq 1} \beta_i^\star \cdot \varphi_i$ will estimate $f(p)$ with an accuracy of $a' + b' \cdot k + c' \cdot \log m$ and a failure probability at most $o(1/\text{poly}(m))$.*

Following the construction in [83] (page 124), let $z := (z_0, z_1, \ldots)$ be the vector of coefficients induced by any solution of the dual program of the previously mentioned linear program. For our purpose, the way in which these coefficients are derived is largely irrelevant. One can show that $|z_\ell| \leq v \cdot n^{c_2}, \forall \ell \geq 0$. Let $t_n := 2n^{-c_1} \log n$ and $\alpha \in (0, 1)$, and define

$$\beta_i := (1 - e^{-t_n \alpha i}) f\left(\frac{(i+1)\alpha}{n}\right) \frac{n}{(i+1)\alpha} + \sum_{\ell=0}^{i} z_\ell (1 - t_n)^\ell \alpha^\ell (1 - \alpha)^{i-\ell} \binom{i}{\ell}.$$

for any $i \leq n$, and $\beta_i := \beta_n$ for $i > n$. The next lemma shows that we can find proper parameters $a, b$, and $c$ to apply Lemma 2 to the above construction. Specifically,

**Lemma 3.** *For any $\alpha \in [1/100, 1)$ and some $a'', b'' \geq 0$ such that $a'' + b''k \leq v$, if $v \leq \log^2 n$ and $c_1, c_2$ satisfy $\alpha c_2 + (3/2 - \alpha)c_1 \leq 1/4$, the two conditions in Lemma 2 hold for the above construction with $m = n/\alpha$, $a' = a'' + \mathcal{O}(n^{-c_1/2} \log^2 n)$, $b' = b''(1 + \mathcal{O}(t_n))$, and $c' = \mathcal{O}(n^{-1/4} \log^3 n)$. Furthermore, for any $i \geq 0$, we have $|\beta_i| \leq \mathcal{O}(n^{\alpha c_2 + (1-\alpha)c_1} \log^3 n)$.*

This lemma differs from the results established in the proof of Proposition 6.19 in [83] only in the applicable range of $\alpha$, where the latter assumes that $\alpha \in [1/2, 1)$. For completeness, we will present a proof of Lemma 3 in Appendix A.

By Lemma 2 and 3, if $v \leq \log^2 n$, given a sample $X^{n/\alpha}$ from any $p \in \Delta_\mathcal{X}$, the linear estimator $\sum_{i \geq 1} \beta_i^\star \cdot \varphi_i$ will estimate $f(p)$ with an accuracy of $a' + b'k + c' \log(n/\alpha) = a'' + \mathcal{O}(n^{-c_1/2} \log^2 n) + b''k(1 + \mathcal{O}(t_n)) + \mathcal{O}(n^{-1/4} \log^4 n) \leq v(1 + \mathcal{O}(t_n)) + \mathcal{O}(n^{-c_1/2} \log^2 n)$ and a failure probability at most $o(1/\text{poly}(n))$. Recall that for fixed $\mathcal{X}, n$, and $f$, the value of $v$ is a constant, thus can be computed without samples. Furthermore according to the last claim in Proposition 6.19 in [83], for $v > \log^2 n$, the estimator that always returns 0 has an error of at most $(1 + o(1))v$. Hence with high probability, the estimator $\hat{f}^\star := \sum_{i \geq 1} (\beta_i^\star \cdot \mathbb{1}_{v \leq \log^2 n}) \cdot \varphi_i$ will estimate $f(p)$ up to an error of $v(1 + o(1)) + \mathcal{O}(t_n \log n)$, for any possible values of $v$.

**Concentration of linear estimators**   Third, we slightly diverge from the previous discussion and study the concentration of general linear estimators.

The *sensitivity* of a property estimator $\hat{f} : \mathcal{X}^* \to \mathbb{R}$ for a given input size $n$ is

$$s_n(\hat{f}) := \max \left\{ f(x^n) - f(y^n) : x^n \text{ and } y^n \text{ differ in one element} \right\},$$

the maximum change in its value when the input sequence is modified at exactly one location. For any $p \in \Delta_\mathcal{X}$ and $X^n \sim p$, the following corollary of the McDiarmid's inequality [60] relates the two-side tail probability of $\hat{f}(X^n)$ to $s_n(\hat{f})$.

**Lemma 4.** *For all $t \geq 0$, we have* $\Pr\left(|\hat{f}(X^n) - \mathbb{E}[\hat{f}(X^n)]| \geq t\right) \leq 2\exp(-2t^2 \cdot (\sqrt{n}s_n(\hat{f}))^{-2})$.

Define $\ell_0 := 0$. The next lemma bounds the sensitivity of a linear estimator $\hat{f} := \sum_{i \geq 1} \ell_i \cdot \varphi_i$ in terms of $\max_{i \geq 1} |\ell_i - \ell_{i-1}|$, the maximum absolute difference between its consecutive coefficients.

**Lemma 5.** *For any $n$ and linear estimator $\hat{f} := \sum_{i \geq 1} \ell_i \cdot \varphi_i$, we have $s_n(\hat{f}) \leq 2\max_{i \geq 1} |\ell_i - \ell_{i-1}|$.*

*Proof.* Let $x^n$ and $y^n$ be two arbitrary sequences over $\mathcal{X}$ that differ in one element. Let $i$ be the index where $x_i \neq y_i$. Then by definition, the following multiplicity equalities hold: $\mu_{x_i}(x^n) = \mu_{x_i}(y^n) + 1$, $\mu_{y_i}(y^n) = \mu_{y_i}(x^n) + 1$, and $\mu_x(x^n) = \mu_x(y^n)$ for $x \in \mathcal{X}$ satisfying $x \neq x_i, y_i$. For simplicity of notation, let $\mu_0 := \mu_{x_i}(x^n)$, $\mu_1 := \mu_{y_i}(y^n)$, and for any $i \geq 1$, let $\hat{f}_i := \ell_{i-1} \cdot \varphi_{i-1} + \ell_i \cdot \varphi_i$.

The first multiplicity equality implies $\varphi_{\mu_0}(x^n) = \varphi_{\mu_0}(y^n) + 1$ and $\varphi_{\mu_0-1}(x^n) = \varphi_{\mu_0-1}(y^n) - 1$. Therefore, we have $\hat{f}_{\mu_0}(x^n) - \hat{f}_{\mu_0}(y^n) = \ell_{\mu_0} - \ell_{\mu_0-1}$. Similarly, the second equality implies $\hat{f}_{\mu_1}(x^n) - \hat{f}_{\mu_1}(y^n) = -\ell_{\mu_1} + \ell_{\mu_1-1}$. The third equality combines these two results and yields

$$\hat{f}(x^n) - \hat{f}(y^n) = \ell_{\mu_0} - \ell_{\mu_0-1} + (-\ell_{\mu_1} + \ell_{\mu_1-1}).$$

Applying the triangle inequality to the right-hand side completes the proof. $\square$

By these two lemmas, we have the following result for the concentration of linear estimators.

**Corollary 1.** *For any $t \geq 0$, $p \in \Delta_{\mathcal{X}}$, and $\hat{f} := \sum_{i \geq 1} \ell_i \cdot \varphi_i$, if $X^n \sim p$, then*

$$\Pr\left(|\hat{f}(X^n) - \mathbb{E}[\hat{f}(X^n)]| \geq t\right) \leq 2\min_{i \geq 1}\exp(-t^2 \cdot (\sqrt{2n}(\ell_i - \ell_{i-1}))^{-2}).$$

**Sensitivity bound** Fourth, we bound the sensitivity of $\hat{f}^\star = \sum_{i \geq 1}(\beta_i^\star \cdot \mathbb{1}_{v \leq \log^2 n}) \cdot \varphi_i$. By Lemma 5, it suffices to consider the absolute difference between consecutive $\beta_i^\star$'s. We assume $v \leq \log^2 n$ and $\alpha \in [1/100, 1)$, and analyze two cases below, depending on whether $i$ is greater than $400n^{c_1}$ or not.

By Lemma 3, for $i \leq 400n^{c_1}$, we have $|\beta_i| \leq \mathcal{O}(n^{\alpha c_2 + (1-\alpha)c_1}\log^3 n)$. Define $\beta_{-1} := 0$. Then,

$$|\beta_{i+1}^\star - \beta_i^\star| = \left|\frac{i+1}{n/\alpha}\beta_i - \frac{i}{n/\alpha}\beta_{i-1}\right| \leq \left|\frac{400n^{c_1}+1}{n/\alpha}\beta_i\right| + \left|\frac{400n^{c_1}}{n/\alpha}\beta_{i-1}\right| \leq \mathcal{O}\left(n^{\alpha c_2 + (2-\alpha)c_1-1}\log^3 n\right).$$

For $i > 400n^{c_1}$, we only need to consider $i < n$ since $\beta_{i+1}^\star = \beta_i^\star$ for all $i \geq n$. Then,

$$
\begin{aligned}
|\beta_{i+1}^\star - \beta_i^\star| &\overset{(a)}{\leq} \left|\sum_{\ell=0}^{i} z_\ell(1-t_n)^\ell \alpha^\ell(1-\alpha)^{i-\ell}\binom{i}{\ell}\frac{(i+1)\alpha}{n}\right| + \left|\sum_{\ell=0}^{i-1} z_\ell(1-t_n)^\ell \alpha^\ell(1-\alpha)^{i-1-\ell}\binom{i-1}{\ell}\frac{i\alpha}{n}\right| \\
&\quad + \left|f\left(\frac{(i+1)\alpha}{n}\right) - f\left(\frac{i\alpha}{n}\right)\right| + \left|e^{-t_n\alpha i}f\left(\frac{(i+1)\alpha}{n}\right)\right| + \left|e^{-t_n\alpha(i-1)}f\left(\frac{i\alpha}{n}\right)\right| \\
&\overset{(b)}{\leq} (n^{c_2}\log^2 n)\left(\left|\sum_{\ell=0}^{i}(1-t_n)^\ell \alpha^\ell(1-\alpha)^{i-\ell}\binom{i}{\ell}\right| + \left|\sum_{\ell=0}^{i-1}(1-t_n)^\ell \alpha^\ell(1-\alpha)^{i-1-\ell}\binom{i-1}{\ell}\right|\right) \\
&\quad + \left|f\left(\frac{(i+1)\alpha}{n}\right) - f\left(\frac{i\alpha}{n}\right)\right| + \left|e^{-t_n\alpha i}f\left(\frac{(i+1)\alpha}{n}\right)\right| + \left|e^{-t_n\alpha(i-1)}f\left(\frac{i\alpha}{n}\right)\right| \\
&\overset{(c)}{\leq} (n^{c_2}\log^2 n)((1-t_n\alpha)^i + (1-t_n\alpha)^{i-1}) + \left|f\left(\frac{(i+1)\alpha}{n}\right) - f\left(\frac{i\alpha}{n}\right)\right| + 2e^{-t_n\alpha(i-1)}/e \\
&\overset{(d)}{\leq} 2(n^{c_2}\log^2 n)\left(1 - \frac{\log n}{50n^{c_1}}\right)^{400n^{c_1}} + \left|f\left(\frac{(i+1)\alpha}{n}\right) - f\left(\frac{i\alpha}{n}\right)\right| + 2n^{-2}/e \\
&\overset{(e)}{=} 2(n^{c_2}\log^2 n)\left(\left(1 - \frac{\log n}{50n^{c_1}}\right)^{\frac{50n^{c_1}}{\log n}}\right)^{8\log n} + \left|f\left(\frac{(i+1)\alpha}{n}\right) - f\left(\frac{i\alpha}{n}\right)\right| + 2n^{-2}/e \\
&\overset{(f)}{\leq} 2n^{-2} + \left|f\left(\frac{(i+1)\alpha}{n}\right) - f\left(\frac{i\alpha}{n}\right)\right|,
\end{aligned}
$$

where (a) follows from the triangle inequality; (b) follows from $i \leq n$, $v \leq \log^2 n$, and $|z_\ell| \leq v \cdot n^{c_2}$ for all $\ell \geq 0$; (c) follows from the binomial theorem and $|f(x)| \leq x|\log x| \leq 1/e$ for $x \in (0, 1]$; (d) follows from $\alpha \geq 1/100$, $i > 400n^{c_1}$, and $t_n = 2n^{-c_1} \log n$; (e) follows from simple algebra; and (f) follows from $c_2 = 1/2 + 6c_1 < 1$ and $(1 - 1/x)^x \leq e^{-1}$ for $x > 1$.

It remains to analyze the second term on the right-hand side.

$$
\begin{aligned}
\left| f\left(\frac{(i+1)\alpha}{n}\right) - f\left(\frac{i\alpha}{n}\right) \right| &\overset{(a)}{=} \frac{(i+1)\alpha}{n} \left| f\left(\frac{(i+1)\alpha}{n}\right) \frac{n}{(i+1)\alpha} - f\left(\frac{i\alpha}{n}\right) \frac{n}{(i+1)\alpha} \right| \\
&\overset{(b)}{=} \frac{(i+1)\alpha}{n} \left| f\left(\frac{(i+1)\alpha}{n}\right) \frac{n}{(i+1)\alpha} - f\left(\frac{i\alpha}{n}\right) \frac{n}{i\alpha} + f\left(\frac{i\alpha}{n}\right) \frac{n}{i(i+1)\alpha} \right| \\
&\overset{(c)}{\leq} \frac{(i+1)\alpha}{n} \left| \log \frac{i+1}{i} \right| + \frac{(i+1)\alpha}{n} \left| \frac{i\alpha}{n} \left( \log\left(\frac{i\alpha}{n}\right) \right) \frac{n}{i(i+1)\alpha} \right| \\
&\overset{(d)}{\leq} \frac{(i+1)\alpha}{n} \frac{1}{i} + \mathcal{O}\left(\frac{\log n}{n}\right) \overset{(e)}{\leq} \mathcal{O}\left(\frac{\log n}{n}\right),
\end{aligned}
$$

where (a), (b) and (e) follows from simple algebra; (c) follows from $|f(x)/x - f(y)/y| \leq |\log(x/y)|$ for all $x, y \in (0, 1]$; (d) follows from $\log(1 + x) \leq x$ for $x \geq 0$ and $x|\log x| \leq 1/e$ for $x \in (0, 1]$.

Consolidating the above inequalities and applying Lemma 5, we get the sensitivity bound

$$
s_n(f^\star) \leq \mathcal{O}\left( n^{\alpha c_2 + (2-\alpha)c_1 - 1} \log^3 n \right).
$$

**Competitiveness of PML**  A property estimator $\hat{f}$ is *profile-based* if there exists a mapping $\hat{g}$ such that $\hat{f}(x^n) = \hat{g}(\varphi(x^n))$ for all $x^n \in \mathcal{X}^*$. The following lemma [2, 5, 29] states that the PML estimator is competitive to other profile-based estimators.

**Lemma 6.** *For any positive real numbers $\varepsilon$ and $\delta$, additive symmetric property $f$, and profile-based estimator $\hat{f}$, the PML-plug-in estimator $f(p_\varphi)$ satisfies*

$$
n_f(f(p_\varphi), 2\varepsilon, \delta \cdot \exp(3\sqrt{n})) \leq n_f(\hat{f}, \varepsilon, \delta).
$$

*For any $\beta$-approximate PML, a similar result holds with $\delta \cdot \exp(3\sqrt{n})$ replaced by $\delta \cdot \exp(3\sqrt{n})/\beta$.*

The factor $\exp(3\sqrt{n})$ directly comes from the well-known result of Hardy and Ramanujan [49] on integer partitions, since there is a bijective mapping from profiles of size $n$ to partitions of integer $n$.

**Final analysis**  Finally, we combine the above results and establish Theorem 1.

Denote by $\tau(n)$ the previous upper bound on $s_n(f^\star)$. Let $p$ be a distribution in $\Delta_\mathcal{X}$ and $X^n \sim p$. Let $\gamma$ be an absolute constant in $(0, 1/4)$. Then by Lemma 4,

$$
\Pr\left( |\hat{f}^\star(X^n) - \mathbb{E}[\hat{f}^\star(X^n)]| \geq 2n^{1-\gamma}\tau(n) \right) \leq 2\exp(-8n^{1-2\gamma}).
$$

Let $\varepsilon > 0$ be an error parameter. Assume there exists an estimator $\hat{f}$ that, when given a length-$\alpha n$ sample from any distribution $p' \in \Delta_\mathcal{X}$, estimates $f(p')$ up to an absolute error $\varepsilon$ with probability at least $2/3$. Then according to the results in the upper- and lower-bound sections, with probability at most $o(1/\mathrm{poly}(n))$, the estimate $\hat{f}^\star(X^n)$ will differ from $f(p)$ by more than $v(1 + o(1)) + \mathcal{O}(n^{-c_1/2} \log^2 n) \leq \varepsilon(1 + o(1)) + \mathcal{O}(n^{-c_1/2} \log^2 n)$. In addition, by the equality $\sum_{i \geq 1} i \cdot \varphi_i(X^n) = n$ and Lemma 3, we surely have $|\hat{f}^\star(X^n)| \leq |\sum_{i \geq 1}(i/m)\beta_{i-1} \cdot \varphi_i(X^n)| \leq \max_{i \geq 0} |\beta_i| \leq \mathcal{O}(n^{\alpha c_2 + (1-\alpha)c_1} \log^3 n)$. Multiplying this bound by $o(1/\mathrm{poly}(n))$ yields a quantity that is negligible comparing to $\mathcal{O}(n^{-c_1/2} \log^2 n)$. Therefore, the absolute bias $|\mathbb{E}[\hat{f}^\star(X^n)] - f(p)|$ is at most $\varepsilon(1 + o(1)) + \mathcal{O}(n^{-c_1/2} \log^2 n)$. The triangle inequality combines this with the tail bound above:

$$
\Pr\left( |\hat{f}(X^n) - f(p)| \geq \varepsilon(1 + o(1)) + \mathcal{O}(n^{-c_1/2} \log^2 n) + 2n^{1-\gamma}\tau(n) \right) \leq 2\exp\left(-8n^{1-2\gamma}\right).
$$

Let $\alpha = 1/4$. For PML and APML estimators, set $(\gamma, c_1)$ to be $(1/4, 1/31)$ and $(0.166, 1/91)$, respectively. Combined, the last inequality and Lemma 6 imply Theorem 1. There is a simple trade-off between $\alpha$ and $c_1$ induced by our proof technique. Specifically, if we increase the value of $c_1$ to achieve a better lower bound on $\varepsilon$, the value of $\alpha$ may need to be reduced accordingly, which enlarges the sample complexity gap between our estimators and the optimal one. For example, reducing $\alpha$ to $1/12$ and $1/22$, we can improve $c_1$ to $1/25$ and $1/20$, respectively, for both PML and APML.

# 6 $\alpha$-Rényi entropy estimation

For any $p \in \Delta_{\mathcal{X}}$ and non-negative $\alpha \neq 1$, the $\alpha$-*Rényi entropy* [74] of $p$ is

$$H_\alpha(p) := \frac{1}{1-\alpha} \log P_\alpha(p) = \frac{1}{1-\alpha} \log \left( \sum_x p(x)^\alpha \right).$$

For $\mathcal{X}$ of finite size $k$ and any $p \in \Delta_{\mathcal{X}}$, it is well-known that $H_\alpha(p) \in [0, \log k]$.

## 6.1 Proof of Theorem 2: $\alpha \in (3/4, 1)$

For $\alpha \in (3/4, 1)$, the following theorem characterizes the performance of the PML-plug-in estimator.

For any distribution $p \in \Delta_{\mathcal{X}}$, error parameter $\varepsilon \in (0, 1)$, and sampling parameter $n$, draw a sample $X^n \sim p$ and denote its profile by $\varphi$. Then for sufficiently large $k$,

**Theorem 2.** *For an $\alpha \in (3/4, 1)$, if $n = \Omega_\alpha(k^{1/\alpha}/(\varepsilon^{1/\alpha} \log k))$,*

$$\Pr\left(|H_\alpha(p_\varphi) - H_\alpha(p)| \geq \varepsilon\right) \leq \exp(-\sqrt{n}).$$

We establish both this theorem and an analogous result for APML in the remaining section. Let $n$ be a sampling parameter and $p \in \Delta_{\mathcal{X}}$ be an unknown distribution. For some $\alpha$-dependent positive constants $c_{\alpha,1}$ and $c_{\alpha,2}$ to be determined later, let $\tau := c_{\alpha,1} \log n$ and $d := c_{\alpha,2} \log n$ be threshold and degree parameters, respectively. Let $N, N'$ be independent Poisson random variables with mean $n$. Consider Poisson sampling with two samples drawn from $p$, first of size $N$ and the second $N'$. Suppressing the sample representations, for each $x \in \mathcal{X}$, we denote by $\mu_x$ and $\mu'_x$ the multiplicities of symbol $x$ in the first and second samples, respectively. Denote by $q(z) := \sum_{m=0}^d a_m z^m$ be the degree-$d$ min-max polynomial approximation of $z^\alpha$ over $[0, 1]$. We consider the following variant of the polynomial-based estimator proposed in [6].

$$\hat{P}_\alpha := \sum_x \left( \sum_{m=0}^d \frac{a_m (2\tau)^{\alpha-m} \mu_x^m}{n^\alpha} \right) \mathbb{1}_{\mu_x \leq 4\tau} \cdot \mathbb{1}_{\mu'_x \leq \tau} + \sum_x \left( \frac{\mu_x}{n} \right)^\alpha \mathbb{1}_{\mu'_x > \tau}.$$

The smaller the value of $\mu'_x$ is, the smaller we expect the value of $p(x)$ to be. In view of this, we denote the first and second components of $\hat{P}_\alpha$ by $\hat{P}_\alpha^{(s)}$ and $\hat{P}_\alpha^{(\ell)}$, and refer to them as small- and large-probability estimators, respectively. Note that our estimator differs from that in [6] only by the additional $\mathbb{1}_{\mu_x \leq 4\tau}$ term, which for sufficiently large $c_{\alpha,1}$, only modifies $\mathbb{E}[\hat{P}_\alpha^{(s)}]$ by at most $n^{-2\alpha}$.

Note that $\mu'$ naturally induces a partition over $\mathcal{X}$. For symbols $x$ with $\mu_x \leq 4\tau$, we denote by

$$P_{a,\mu'}^{(s)}(p) := \sum_{x:\mu_x \leq 4\tau} p(x)^\alpha$$

the small-probability power sum. Analogously, for symbols $x$ with $\mu_x > 4\tau$, we denote by

$$P_{a,\mu'}^{(\ell)}(p) := \sum_{x:\mu_x > 4\tau} p(x)^\alpha$$

the large-probability power sum. These are random properties with non-trivial variances and are hard to be analyzed. To address this, we apply an "expectation trick" and denote by $P_a^{(s)}(p) := \mathbb{E}[P_{a,\mu'}^{(s)}(p)]$ and $P_a^{(\ell)}(p) := \mathbb{E}[P_{a,\mu'}^{(\ell)}(p)]$ their expected values, both of which are additive symmetric properties.

Let $\varepsilon$ be a given error parameter and $n = \Omega_\alpha(k^{1/\alpha}/(\varepsilon^{1/\alpha} \log k))$ be a sampling parameter. First we consider the small probability estimator. By the results in [6], for sufficiently large $c_{\alpha,1}$, the bias of $\hat{P}_\alpha^{(s)}$ in estimating $P_\alpha^{(s)}(p)$ satisfies

$$|\mathbb{E}[\hat{P}_\alpha^{(s)}] - P_\alpha^{(s)}(p)| \leq \mathcal{O}_\alpha(1) \cdot P_\alpha(p) \left( \frac{k}{n \log n} \right)^\alpha + n^{-\alpha} \leq \varepsilon P_\alpha(p),$$

where we have used $n^{-\alpha} = \mathcal{O}_\alpha(\varepsilon k^{-1}(\log k)^\alpha) \leq \varepsilon P_\alpha(p)$. To show concentration, we bound the sensitivity of estimator $\hat{P}_\alpha^{(s)}$. For $m \geq 0$, we can bound the coefficients of $q(x)$ as follows.

$$|a_m| = \mathcal{O}_\alpha((\sqrt{2}+1)^d) = \mathcal{O}_\alpha(n^{c_{\alpha,2}}).$$

Therefore by definition, changing one point in the sample changes the value of $\hat{P}_\alpha^{(s)}$ by at most

$$2\left(\sum_{m=0}^d \frac{|a_m|(2\tau)^{\alpha-m}(4\tau)^m}{n^\alpha}\right) \le \sum_{m=0}^d \frac{|a_m|(2\tau)^\alpha 2^{m+1}}{n^\alpha} = \mathcal{O}_\alpha\left(n^{2c_{\alpha,2}-\alpha}(\log n)^\alpha\right).$$

Let $\lambda \in (0, 1/4)$ be an arbitrary absolute constant. For sufficiently small $c_{\alpha,2}$, the right-hand side is at most $\mathcal{O}_\alpha\left(n^{\lambda-\alpha}\right)$. The McDiarmid's inequality together with the concentration of Poisson random variables implies that for all $\varepsilon \ge 0$,

$$\Pr\left(|\hat{P}_\alpha^{(s)} - \mathbb{E}[\hat{P}_\alpha^{(s)}]| \ge \varepsilon P_\alpha(p)\right) \le 2\exp(-\Omega_\alpha(\varepsilon^2 P_\alpha^2(p)n^{2\alpha-1-2\lambda})).$$

Note that $n = \Omega_\alpha(k^{1/\alpha}/(\varepsilon^{1/\alpha}\log k))$ and $P_\alpha(p) \ge 1$, which follows from the fact that $z^\alpha$ is a concave function over $[0,1]$ for $\alpha \in (0,1)$. Hence we obtain

$$\Pr\left(|\hat{P}_\alpha^{(s)} - \mathbb{E}[\hat{P}_\alpha^{(s)}]| \ge \varepsilon P_\alpha(p)\right) \le 3\exp\left(-\Omega_\alpha\left(\varepsilon^2 n^{2\alpha-1-2\lambda}\right)\right).$$

For $\alpha > 3/4$, we can set $\lambda = (4\alpha - 3)/8$. Direct calculation shows that for sufficiently large $k$, the right-hand side is no more than $\exp(-8\sqrt{n})$. Analogously, we can show that for $\alpha > 5/6$, the probability bound can be improved to $\exp(-\Theta(n^{2/3}))$.

Second, we consider the large probability estimator. To begin with, we set $n = \Theta_\alpha(k^{1/3})$. By the results in [6], for sufficiently large $c_{\alpha,1}$, the bias of $\hat{P}_\alpha^{(\ell)}$ in estimating $P_\alpha^{(\ell)}(p)$ satisfies

$$|\mathbb{E}[\hat{P}_\alpha^{(\ell)}] - P_\alpha^{(\ell)}(p)| \le \mathcal{O}_\alpha\left(\frac{P_\alpha(p)}{\tau}\right) + \frac{1}{n^\alpha},$$

which, for sufficiently large $k$, is at most $\varepsilon P_\alpha(p)$. Under the same conditions, the variance of $\hat{P}_\alpha^{(\ell)}$ is at most

$$\text{Var}(\hat{P}_\alpha^{(\ell)}) \le \mathcal{O}_\alpha\left(\sum_x \frac{p(x)^{2\alpha}}{\tau}\right) + \frac{1}{n^{2\alpha}} \le \frac{(\varepsilon P_\alpha(p))^2}{3}.$$

Then, the Chebyshev's inequality yields

$$\Pr\left(|\mathbb{E}[\hat{P}_\alpha^{(\ell)}] - \hat{P}_\alpha^{(\ell)}| \ge \varepsilon P_\alpha(p)\right) \le \frac{1}{3}.$$

The triangle inequality combines this tail bound with the above bias bound and implies

$$\Pr\left(|P_\alpha^{(\ell)}(p) - \hat{P}_\alpha^{(\ell)}| \ge 2\varepsilon P_\alpha(p)\right) \le \frac{1}{3}.$$

Therefore, utilizing the median trick and $\alpha < 1$, we can construct another estimator $\hat{P}_\alpha^{(\ell,1)}$ that takes a sample of size $n = \Omega_\alpha(k^{1/\alpha}/(\varepsilon^{1/\alpha}\log k))$, and satisfies

$$\Pr\left(|P_\alpha^{(\ell)}(p) - \hat{P}_\alpha^{(\ell,1)}| \ge 2\varepsilon P_\alpha(p)\right) \le 2\exp(-\Omega_\alpha(n/k^{1/3}))) \le 2\exp(-\Theta(n^{2/3})).$$

Recall that $P_\alpha(p) = P_\alpha^{(s)}(p) + P_\alpha^{(\ell)}(p)$. By the union bound and the triangle inequality, under Poisson sampling with parameter $n = \Theta_\alpha(k^{1/\alpha}/(\varepsilon^{1/\alpha}\log k))$,

$$\Pr\left(|P_\alpha(p) - (\hat{P}_\alpha^{(s)} + \hat{P}_\alpha^{(\ell,1)})| \ge 4\varepsilon P_\alpha(p)\right) \le \exp(-8\sqrt{n}).$$

Since both $N$ and $N'$ are Poisson random variables with mean $n$, we must have $N + N' \sim \text{Poi}(2n)$, implying that $\Pr(N + N' = 2n) = e^{-2n}(2n)^{2n}/(2n)!$. A variant of the well-known Stirling's formula states that $m! \ge em^{m+1/2}e^{-m}$ for all positive integers $m$. We obtain $\Pr(N + N' = 2n) \ge e^{-2n}(2n)^{2n} \cdot (e(2n)^{2n+1/2}e^{-2n})^{-1} \ge 1/(e\sqrt{2n}) > 1/(4n)$. Hence, under fixed sampling with a sample size of $2n$, the estimator $\hat{P}_\alpha^{(1)} := (\hat{P}_\alpha^{(s)} + \hat{P}_\alpha^{(\ell,1)})$ satisfies

$$\Pr\left(|P_\alpha(p) - \hat{P}_\alpha^{(1)}| \ge 4\varepsilon P_\alpha(p)\right) \le 4n\exp(-8\sqrt{n}).$$

Replacing $n$ with $n/2$ and $\varepsilon$ with $\varepsilon/4$, the *sufficiency of profiles* [6] implies the existence of a profile-based estimator $\hat{P}_\alpha^\star$ such that for any $p \in \Delta_\mathcal{X}$,

$$\Pr_{X^n \sim p}\left(|P_\alpha(p) - \hat{P}_\alpha^\star(X^n)| \geq \varepsilon P_\alpha(p)\right) \leq 2n\exp(-4\sqrt{2n}) < \exp(-4\sqrt{n}).$$

Let $\delta$ denote the quantity on the right-hand side. For any $x^n$ with profile $\varphi$ satisfying both $p(\varphi) > \delta$, we must have $|\hat{P}_\alpha^\star(x^n) - P_\alpha(p)| \leq \varepsilon P_\alpha(p)$. By definition, we also have $p_\varphi(\varphi) \geq p(\varphi) > \delta$ and hence $|\hat{P}_\alpha^\star(x^n) - P_\alpha(p_\varphi)| \leq \varepsilon P_\alpha(p_\varphi)$. For any $\varepsilon \in (0, 1/2)$, simple algebra combines the two property inequalities and yields

$$|P_\alpha(p) - P_\alpha(p_\varphi)| \leq 2\varepsilon P_\alpha(p).$$

On the other hand, for a sample $X^n \sim p$ with profile $\varphi'$, the probability that we have $p(\varphi') \leq \delta$ is at most $\delta$ times the cardinality of the set $\Phi^n := \{\varphi(x^n) : x^n \in \mathcal{X}^n\}$. The latter quantity corresponds to the number of integer partitions of $n$, which, by the well-known result of Hardy and Ramanujan [49], is at most $\exp(3\sqrt{n})$. Hence, the probability that $p(\varphi') \leq \delta$ is upper bounded by $\exp(-\sqrt{n})$. To conclude, we have shown that

$$\Pr\left(|P_\alpha(p) - P_\alpha(p_\varphi)| \geq 2\varepsilon P_\alpha(p)\right) \leq \exp(-\sqrt{n}).$$

In terms of Rényi entropy values, applying the inequality $e^z - 1 \geq 1 - e^{-z} \geq z/2$ for all $z \geq 0$, we establish that for $\alpha > 3/4$ and $n = \Omega_\alpha(k/(\varepsilon^{1/\alpha}\log k))$,

$$\Pr\left(|H_\alpha(p) - H_\alpha(p_\varphi)| \geq \varepsilon\right) = \Pr\left(P_\alpha(p_\varphi)e^{-(\alpha-1)\varepsilon} \leq P_\alpha(p) \leq P_\alpha(p_\varphi)e^{(\alpha-1)\varepsilon}\right) \leq \exp(-\sqrt{n}).$$

## 6.2 Proof of Theorem 3: Non-integer $\alpha > 1$

The proof of the following theorem is essentially the same as that shown in the previous section. However, for completeness, we still include a full-length proof.

For any distribution $p \in \Delta_\mathcal{X}$, error parameter $\varepsilon \in (0, 1)$, absolute constant $\lambda \in (0, 0.1)$, and sampling parameter $n$, draw a sample $X^n \sim p$ and denote its profile by $\varphi$. Then for sufficiently large $k$,

**Theorem 3.** *For a non-integer $\alpha > 1$, if $n = \Omega_\alpha(k/(\varepsilon^{1/\alpha}\log k))$,*

$$\Pr\left(|H_\alpha(p_\varphi) - H_\alpha(p)| \geq \varepsilon\right) \leq \exp(-n^{1-\lambda}).$$

We establish this theorem in the remaining section. Let $n$ be a sampling parameter and $p \in \Delta_\mathcal{X}$ be an unknown distribution. For some $\alpha$-dependent positive constants $c_{\alpha,1}$ and $c_{\alpha,2}$ to be determined later, let $\tau := c_{\alpha,1}\log n$ and $d := c_{\alpha,2}\log n$ be threshold and degree parameters, respectively. Let $N, N'$ be independent Poisson random variables with mean $n$. Consider Poisson sampling with two samples drawn from $p$, first of size $N$ and the second $N'$. Suppressing the sample representations, for each $x \in \mathcal{X}$, we denote by $\mu_x$ and $\mu'_x$ the multiplicities of symbol $x$ in the first and second samples, respectively. Denote by $q(z) := \sum_{m=0}^d a_m z^m$ be the degree-$d$ min-max polynomial approximation of $z^a$ over $[0, 1]$. We consider the following variant of the estimator proposed in [6].

$$\hat{P}_\alpha := \sum_x \left(\sum_{m=0}^d \frac{a_m(2\tau)^{\alpha-m}\mu_x^m}{n^\alpha}\right)\mathbb{1}_{\mu_x \leq 4\tau}\cdot\mathbb{1}_{\mu'_x \leq \tau} + \sum_x \left(\frac{\mu_x}{n}\right)^\alpha \mathbb{1}_{\mu'_x > \tau}.$$

The smaller the value of $\mu'_x$ is, the smaller we expect the value of $p(x)$ to be. In view of this, we denote the first and second components of $\hat{P}_\alpha$ by $\hat{P}_\alpha^{(s)}$ and $\hat{P}_\alpha^{(\ell)}$, and refer to them as small- and large-probability estimators, respectively. Note that our estimator differs from that in [6] only by the additional $\mathbb{1}_{\mu_y \leq 4\tau}$ term, which for sufficiently large $c_{\alpha,1}$, only modifies $\mathbb{E}[\hat{P}_\alpha^{(s)}]$ by at most $n^{-2\alpha}$.

Note that $\mu'$ naturally induces a partition over $\mathcal{X}$. For symbols $x$ with $\mu_x \leq 4\tau$, we denote by

$$P_{a,\mu'}^{(s)}(p) := \sum_{x:\mu_x \leq 4\tau} p(x)^\alpha$$

the small-probability power sum. Analogously, for symbols $x$ with $\mu_x > 4\tau$, we denote by

$$P_{a,\mu'}^{(\ell)}(p) := \sum_{x:\mu_x > 4\tau} p(x)^\alpha$$

the large-probability power sum. These are random properties with non-trivial variances and are hard to be analyzed. To address this, we apply an "expectation trick" and denote by $P_a^{(s)}(p) := \mathbb{E}[P_{a,\mu'}^{(s)}(p)]$ and $P_a^{(\ell)}(p) := \mathbb{E}[P_{a,\mu'}^{(\ell)}(p)]$ their expected values, both of which are additive symmetric properties.

Let $\varepsilon$ be a given error parameter and $n = \Omega_\alpha(k/(\varepsilon^{1/\alpha}\log k))$ be a sampling parameter. First we consider the small probability estimator. By the results in [6], for sufficiently large $c_{\alpha,1}$, the bias of $\hat{P}_\alpha^{(s)}$ in estimating $P_\alpha^{(s)}(p)$ satisfies

$$|\mathbb{E}[\hat{P}_\alpha^{(s)}] - P_\alpha^{(s)}(p)| \leq \mathcal{O}_\alpha(1) \cdot P_\alpha(p)\left(\frac{k}{n\log n}\right)^\alpha + n^{-\alpha} \leq \varepsilon P_\alpha(p),$$

where we have used $n^{-\alpha} = \mathcal{O}_\alpha(\varepsilon k^{-\alpha}(\log k)^\alpha) \leq \varepsilon P_\alpha(p)$. To show concentration, we bound the sensitivity of estimator $\hat{P}_\alpha^{(s)}$. For $m \geq 0$, we can bound the coefficients of $q(x)$ as follows.

$$|a_m| \leq \mathcal{O}_\alpha((\sqrt{2}+1)^d) = \mathcal{O}_\alpha(n^{c_{\alpha,2}}).$$

Therefore by definition, changing one point in the sample changes the value of $\hat{P}_\alpha^{(s)}$ by at most

$$2\left(\sum_{m=0}^d \frac{|a_m|(2\tau)^{\alpha-m}(4\tau)^m}{n^\alpha}\right) \leq \sum_{m=0}^d \frac{|a_m|(2\tau)^\alpha 2^{m+1}}{n^\alpha} \leq \mathcal{O}_\alpha\left(n^{2c_{\alpha,2}-\alpha}(\log n)^\alpha\right).$$

Let $\lambda \in (0, 1/4)$ be an arbitrary absolute constant. For sufficiently small $c_{\alpha,2}$, the right-hand side is at most $\mathcal{O}_\alpha\left(n^{\lambda-\alpha}\right)$. The McDiarmid's inequality together with the concentration of Poisson random variables implies that for all $\varepsilon \geq 0$,

$$\Pr\left(|\hat{P}_\alpha^{(s)} - \mathbb{E}[\hat{P}_\alpha^{(s)}]| \geq \varepsilon P_\alpha(p)\right) \leq 2\exp(-\Omega_\alpha(\varepsilon^2 P_\alpha^2(p)n^{2\alpha-1-2\lambda})).$$

Note that $n = \Omega_\alpha(k/(\varepsilon^{1/\alpha}\log k))$ and $P_\alpha(p) \geq k^{1-\alpha}$. Hence we obtain

$$\Pr\left(|\hat{P}_\alpha^{(s)} - \mathbb{E}[\hat{P}_\alpha^{(s)}]| \geq \varepsilon P_\alpha(p)\right) \leq 3\exp\left(-\Omega_\alpha(\varepsilon^2 k^{2-2\alpha}n^{2\alpha-1-2\lambda})\right).$$

By simple algebra, for sufficiently large $k$, the right-hand side is at most $\exp(-n^{1-3\lambda})$.

Second, we consider the large probability estimator. To begin with, we set $n = \Theta_\alpha(k^\lambda)$. By the results in [6], for sufficiently large $c_{\alpha,1}$, the bias of $\hat{P}_\alpha^{(\ell)}$ in estimating $P_\alpha^{(\ell)}(p)$ satisfies

$$|\mathbb{E}[\hat{P}_\alpha^{(\ell)}] - P_\alpha^{(\ell)}(p)| \leq \mathcal{O}_\alpha\left(\frac{P_\alpha(p)}{\tau}\right) + \frac{1}{n^{4\alpha}},$$

which, for sufficiently large $k$, is at most $\varepsilon P_\alpha(p)$. Under the same conditions, the variance of $\hat{P}_\alpha^{(\ell)}$ is at most

$$\mathrm{Var}(\hat{P}_\alpha^{(\ell)}) \leq \mathcal{O}_\alpha\left(\sum_x \frac{p(x)^{2\alpha}}{\tau}\right) + \frac{1}{n^{8\alpha}} \leq \frac{(\varepsilon P_\alpha(p))^2}{3}.$$

Then, the Chebyshev's inequality yields

$$\Pr\left(|\mathbb{E}[\hat{P}_\alpha^{(\ell)}] - \hat{P}_\alpha^{(\ell)}| \geq \varepsilon P_\alpha(p)\right) \leq \frac{1}{3}.$$

The triangle inequality combines this tail bound with the above bias bound and implies

$$\Pr\left(|P_\alpha^{(\ell)}(p) - \hat{P}_\alpha^{(\ell)}| \geq 2\varepsilon P_\alpha(p)\right) \leq \frac{1}{3}.$$

Therefore, utilizing the median trick, we can construct another estimator $\hat{P}_\alpha^{(\ell,1)}$ that takes a sample of size $n = \Omega_\alpha(k/(\varepsilon^{1/\alpha}\log k))$, and for sufficiently large $k$, satisfies

$$\Pr\left(|P_\alpha^{(\ell)}(p) - \hat{P}_\alpha^{(\ell,1)}| \geq 2\varepsilon P_\alpha(p)\right) \leq 2\exp(-\Omega_\alpha(n/k^\lambda)) \leq \exp(-n^{1-2\lambda}).$$

Recall that $P_\alpha(p) = P_\alpha^{(s)}(p) + P_\alpha^{(\ell)}(p)$. By the union bound and the triangle inequality, under Poisson sampling with parameter $n = \Omega_\alpha(k/(\varepsilon^{1/\alpha} \log k))$,

$$\Pr\left(|P_\alpha(p) - (\hat{P}_\alpha^{(s)} + \hat{P}_\alpha^{(\ell,1)})| \geq 4\varepsilon P_\alpha(p)\right) \leq \exp(-n^{1-3\lambda}).$$

Since both $N$ and $N'$ are Poisson random variables with mean $n$, we must have $N + N' \sim \text{Poi}(2n)$, implying that $\Pr(N + N' = 2n) = e^{-2n}(2n)^{2n}/(2n)!$. A variant of the well-known Stirling's formula states that $m! \geq e m^{m+1/2} e^{-m}$ for all positive integers $m$. We obtain $\Pr(N + N' = 2n) \geq e^{-2n}(2n)^{2n} \cdot (e(2n)^{2n+1/2}e^{-2n})^{-1} \geq 1/(e\sqrt{2n}) > 1/(4n)$. Hence, under fixed sampling with a sample size of $2n$, the estimator $\hat{P}_\alpha^{(1)} := (\hat{P}_\alpha^{(s)} + \hat{P}_\alpha^{(\ell,1)})$ satisfies

$$\Pr\left(|P_\alpha(p) - \hat{P}_\alpha^{(1)}| \geq 4\varepsilon P_\alpha(p)\right) \leq 4n \exp(-n^{1-3\lambda}).$$

Replacing $\varepsilon$ with $\varepsilon/4$ and $\lambda$ with $\lambda/5$, the sufficiency of profiles implies the existence of a profile-based estimator $\hat{P}_\alpha^\star$ such that for sufficiently large $k$ and any $p \in \Delta_\mathcal{X}$,

$$\Pr_{X^n \sim p}\left(|P_\alpha(p) - \hat{P}_\alpha^\star(X^n)| \geq \varepsilon P_\alpha(p)\right) \leq 4n \exp(-n^{1-3\lambda/5}) < \exp(-n^{1-4\lambda/5}).$$

Let $\delta$ denote the quantity on the right-hand side. For any $x^n$ with profile $\varphi$ satisfying both $p(\varphi) > \delta$, we must have $|\hat{P}_\alpha^\star(x^n) - P_\alpha(p)| \leq \varepsilon P_\alpha(p)$. By definition, we also have $p_\varphi(\varphi) \geq p(\varphi) > \delta$ and hence $|\hat{P}_\alpha^\star(x^n) - P_\alpha(p_\varphi)| \leq \varepsilon P_\alpha(p_\varphi)$. For any $\varepsilon \in (0, 1/2)$, simple algebra combines the two property inequalities and yields

$$|P_\alpha(p) - P_\alpha(p_\varphi)| \leq 2\varepsilon P_\alpha(p).$$

On the other hand, for a sample $X^n \sim p$ with profile $\varphi'$, the probability that we have $p(\varphi') \leq \delta$ is at most $\delta$ times the cardinality of the set $\Phi^n := \{\varphi(x^n) : x^n \in \mathcal{X}^n\}$. The latter quantity corresponds to the number of integer partitions of $n$, which, by the well-known result of Hardy and Ramanujan [49], is at most $\exp(3\sqrt{n})$. Hence, the probability that $p(\varphi') \leq \delta$ is upper bounded by $\exp(-n^{1-\lambda})$. To conclude, we have shown that

$$\Pr\left(|P_\alpha(p) - P_\alpha(p_\varphi)| \geq 2\varepsilon P_\alpha(p)\right) \leq \exp(-n^{1-\lambda}).$$

In terms of Rényi entropy values, applying the inequality $e^z - 1 \geq 1 - e^{-z} \geq z/2$ for all $z \geq 0$, we establish that for $n = \Omega_\alpha(k/(\varepsilon^{1/\alpha} \log k))$,

$$\Pr\left(|H_\alpha(p) - H_\alpha(p_\varphi)| \geq \varepsilon\right) = \Pr\left(P_\alpha(p_\varphi)e^{-(\alpha-1)\varepsilon} \leq P_\alpha(p) \leq P_\alpha(p_\varphi)e^{(\alpha-1)\varepsilon}\right) \leq \exp(-n^{1-\lambda}).$$

## 6.3 Proof of Theorem 4: Integer $\alpha > 1$

For an integer $\alpha > 1$, the following theorem characterizes the performance of the PML-plug-in estimator. For any $p \in \Delta_\mathcal{X}$, $\varepsilon \in (0, 1)$, and a sample $X^n \sim p$ with profile $\varphi$,

**Theorem 4.** *If* $n = \Omega_\alpha(k^{1-1/\alpha}(\varepsilon^2|\log \varepsilon|)^{-(1+\alpha)})$ *and* $H_\alpha(p) \leq (\log n)/4$,

$$\Pr(|H_\alpha(p_\varphi) - H_\alpha(p)| \geq \varepsilon) \leq 1/3.$$

Due to the lower bounds in [6], for all possible values of $\alpha$, the sample complexity of the PML plug-in estimator has the optimal dependency in $k$. The remaining section is devoted to proving the above theorem. Note that estimating the Rényi entropy $H_\alpha(p)$ to an additive error is equivalent to estimating the power sum $P_\alpha(p)$ to a corresponding multiplicative error. Given this fact, we consider the estimator $\hat{P}_\alpha$ in [6] that maps each sequence $x^n \in \mathcal{X}^*$ to

$$\hat{P}_\alpha(x^n) := \sum_x \frac{\mu_x(x^n)^{\underline{\alpha}}}{n^{\underline{\alpha}}},$$

where for any real number $z$, the expression $z^{\underline{\alpha}}$ denotes the falling factorial of $z$ to the power $\alpha$. For a sample $X^n \sim p$, we have $\mathbb{E}[\hat{P}_\alpha(X^n)] = P_\alpha(p)$. The following lemma [63, 6] states that $\hat{P}_\alpha(X^n)$ often estimates $P_\alpha(p)$ to a small multiplicative error when $n$ is large.

**Lemma 7.** *Under the above conditions, for any $\varepsilon, n > 0$,*

$$\Pr\left(|\hat{P}_\alpha(X^n) - P_\alpha(p)| \geq \varepsilon P_\alpha(p)\right) = \mathcal{O}_\alpha(\varepsilon^{-2} n^{-1} (P_\alpha(p))^{-1/\alpha}).$$

*For sufficiently large $n = \Omega_\alpha(k^{(\alpha-1)/\alpha})$, this inequality together with $P_\alpha(p) \leq k^{1-\alpha}$ implies that*

$$\Pr\left(|\hat{P}_\alpha(X^n) - P_\alpha(p)| \geq \frac{1}{2} \cdot P_\alpha(p)\right) \leq \frac{1}{4}.$$

The following corollary is a consequence of the above lemma, the sufficiency of profiles, and the standard median trick.

**Corollary 2.** *Under the above conditions, there is an estimator $\hat{P}_\alpha^\star$ such that for any $\varepsilon, n > 0$,*

$$\Pr\left(|\hat{P}_\alpha^\star(X^n) - P_\alpha(p)| \geq \varepsilon P_\alpha(p)\right) \leq 2\exp\left(-\Omega_\alpha(\varepsilon^2 n(P_\alpha(p))^{1/\alpha})\right).$$

*In addition, the estimator $\hat{P}_\alpha^\star$ is profile-based.*

For simplicity, suppress $X^n$ in $p_\mu(X^n)$. Since the profile probability $p(\varphi)$ is invariant to symbol permutation, for our purpose, we can assume that $p_\mu(y) \leq p_\mu(z)$ iff $p_\varphi(x) \leq p_\varphi(y)$, for all $x, y \in \mathcal{X}$. Under this assumption, the following lemma [67, 8] relates $p_\varphi$ to $p_\mu$.

**Lemma 8.** *For a distribution $p$ and sample $X^n \sim p$ with profile $\varphi$,*

$$\Pr\left(\max_x |p_\varphi(x) - p_\mu(x)| > \frac{2\log n}{n^{1/4}}\right) = \mathcal{O}\left(\frac{1}{n}\right).$$

Consider $\varepsilon \in (0, 1/2)$ and $x^n$ satisfying $|\hat{P}_\alpha^\star(x^n) - P_\alpha(p)| \leq \varepsilon P_\alpha(p)$. If we further have $P_\alpha(p) \geq 2(n^{1/4}(4\log n)^{-1})^{1-\alpha}$ and $\max_y |p_\varphi(y) - p_\mu(y)| \leq 2(\log n)n^{-1/4}$, then,

$$\frac{P_\alpha(p)}{2} \overset{(a)}{\leq} \hat{P}_\alpha(x^n) \overset{(b)}{\leq} P_\alpha(p_\mu) \overset{(c)}{\leq} 2^{1+\alpha} P_\alpha(p_\varphi),$$

where $(a)$ follows from the above assumptions; $(b)$ follows from $A^{\underline{B}} \leq A^B$ for any $A, B \geq 0$; and $(c)$ follows from the reasoning below.

- Let $S$ denote the the collection of symbols $x$ such that $p_\mu(x) \leq 4(\log n)n^{-1/4}$. Then a convexity argument yields $\sum_{x \in S} (p_\mu(x))^\alpha \leq (n^{1/4}(4\log n)^{-1})^{1-\alpha}$.

- Using $(a)$, $(b)$, and $P_\alpha(p) \geq 4(n^{1/4}(4\log n)^{-1})^{1-\alpha}$, we immediately obtain $P_\alpha(p_\mu) \geq 2(n^{1/4}(4\log n)^{-1})^{1-\alpha}$ and thus $2\sum_{x \in S} (p_\mu(x))^\alpha \leq P_\alpha(p_\mu) \leq 2\sum_{x \notin S} (p_\mu(x))^\alpha$.

- For any symbol $x \notin S$, we have $p_\mu(x) > 4(\log n)n^{-1/4}$. This together with the assumption that $\max_x |p_\varphi(x) - p_\mu(x)| \leq 2(\log n)n^{-1/4}$ implies $p_\mu(x) \leq 2p_\varphi(x)$.

- Therefore, the inequality $\sum_{x \notin S} (p_\mu(x))^\alpha \leq 2^\alpha \sum_{x \notin S} (p_\varphi(x))^\alpha \leq 2^\alpha P_\alpha(p_\varphi)$ holds.

- Consequently, we establish $P_\alpha(p_\mu(x)) \leq 2\sum_{x \notin S} (p_\mu(x))^\alpha \leq 2^{1+\alpha} P_\alpha(p_\varphi)$.

By the inequality $P_\alpha(p)/2 \leq 2^{1+\alpha} P_\alpha(p_\varphi)$ and Corollary 2, if $|\hat{P}_\alpha^\star(x^n) - P_\alpha(p_\varphi)| \geq \varepsilon P_\alpha(p_\varphi)$,

$$p_\varphi(\varphi) \leq 2\exp\left(-\Omega_\alpha(\varepsilon^2 n(P_\alpha(p_\varphi))^{1/\alpha})\right) \leq 2\exp\left(-\Omega_\alpha(\varepsilon^2 n(P_\alpha(p))^{1/\alpha})\right).$$

Let $\delta_p$ denote the quantity on the right-hand side. If we further have $p(\varphi) > \delta_p$, then by definition, $p_\varphi(\varphi) \geq p(\varphi) > \delta_p$. Hence for any $x^n$ with profile $\varphi$ satisfying both $p(\varphi) > \delta_p$ and $|\hat{P}_\alpha^\star(x^n) - P_\alpha(p)| \leq \varepsilon P_\alpha(p)$, we must have $|\hat{P}_\alpha^\star(x^n) - P_\alpha(p_\varphi)| \leq \varepsilon P_\alpha(p_\varphi)$. Simple algebra combines the last two inequalities and yields

$$|P_\alpha(p) - P_\alpha(p_\varphi)| \leq 4\varepsilon P_\alpha(p).$$

On the other hand, for a sample $X^n \sim p$ with profile $\varphi'$, the probability that we have both $p(\varphi') \leq \delta_p$ and $|\hat{P}_\alpha^\star(X^n) - P_\alpha(p)| \leq \varepsilon P_\alpha(p)$ is at most $\delta_p$ times the cardinality of the set $\Phi_{\alpha,\varepsilon}^n(p) := \{\varphi(x^n) :$

$x^n \in \mathcal{X}^n$ and $|\hat{P}_\alpha^\star(x^n) - P_\alpha(p)| \le \varepsilon P_\alpha(p)\}$. Below we complete this argument by finding a tight upper bound on $|\Phi_{\alpha,\varepsilon}^n(p)|$ in terms of its parameters.

For any sequence $x^n$ such that $\varphi(x^n) \in \Phi_{\alpha,\varepsilon}^n(p)$, let $N_\varphi(x^n)$ denote the number of prevalences $\varphi_j(x^n)$ that are non-zero. Then by definition, we obtain

$$\sum_{j=0}^{N_\varphi(x^n)} \frac{j^{\underline{\alpha}}}{n^{\underline{\alpha}}} \le \sum_j \frac{j^{\underline{\alpha}}}{n^{\underline{\alpha}}} \cdot \varphi_j(x^n) = \hat{P}_\alpha^\star(x^n) \le \frac{3}{2} P_\alpha(p).$$

Using the standard falling-factorial identity $((j+1)^{\underline{1+\alpha}} - j^{\underline{1+\alpha}})/(1+\alpha) = j^{\underline{\alpha}}$, we can further simplify the expression on the left-hand side:

$$\sum_{j=0}^{N_\varphi(x^n)} \frac{j^{\underline{\alpha}}}{n^{\underline{\alpha}}} = \frac{(N_\varphi(x^n)+1)^{\underline{1+\alpha}}}{(1+\alpha)n^{\underline{\alpha}}}.$$

This together with the inequality above yields $N_\varphi(x^n) \le T_\alpha^n(p) := (3(1+\alpha)n^{\underline{\alpha}} \cdot P_\alpha(p)/2)^{1/(1+\alpha)}$. Further note that each prevalence in $\varphi(x^n) = (\varphi_1(x^n), \ldots, \varphi_n(x^n))$ can only take values in $\lceil n \rfloor := \{0, 1, \ldots, n\}$. Therefore, $|\Phi_{\alpha,\varepsilon}^n(p)|$ is at most the number of $T_\alpha^n(p)$-sparse vectors over $\lceil n \rfloor^n$, which admits the following upper bound

$$\binom{n}{T_\alpha^n(p)} |\lceil n \rfloor|^{T_\alpha^n(p)} \le (n+1)^{2T_\alpha^n(p)}.$$

Therefore, for $\delta_p \cdot |\Phi_{\alpha,\varepsilon}^n(p)|$ to be small, it suffices to have

$$\Omega_\alpha(\varepsilon^2 n (P_\alpha(p))^{1/\alpha}) \gg 2T_\alpha^n(p) \log(n+1) = 2(3(1+\alpha)n^{\underline{\alpha}} \cdot P_\alpha(p)/2)^{1/(1+\alpha)} \log(n+1),$$

which in turn simplifies to

$$\varepsilon^2 n^{1/(1+\alpha)} (P_\alpha(p))^{1/(\alpha(1+\alpha))} \gg \Theta_\alpha(\log n).$$

Following this and $P_\alpha(p) \ge 4(n^{1/4}(4\log n)^{-1})^{1-\alpha}$, we obtain the following lower bound on $n$.

$$n \gg \Theta_\alpha((\varepsilon^2|\log\varepsilon|)^{-(1+\alpha)}(P_\alpha(p))^{-1/\alpha}).$$

In this case, the probability bound $\delta_p \cdot |\Phi_{\alpha,\varepsilon}^n(p)|$ is no larger than $1/6$.

Finally, let $C$ denote the collection of sequences $x^n$ with profile $\varphi$ that do not satisfy $|\hat{P}_\alpha^\star(x^n) - P_\alpha(p)| \le \varepsilon P_\alpha(p)$ or $\max_x |p_\varphi(x) - \mu_x(x^n)/n| \le 2(\log n)n^{-1/4}$. By Corollary 2, Lemma 9, and the union bound,

$$\Pr_{X^n \sim p}(X^n \in C) \le 2\exp\left(-\Omega_\alpha(\varepsilon^2 n(P_\alpha(p))^{1/\alpha})\right) + \mathcal{O}\left(\frac{1}{n}\right).$$

For $n$ satisfying the lower-bound inequality above, the right-hand side is again no larger than $1/6$. This completes the proof of the theorem.

## 7 Sorted distribution estimation

### 7.1 Sorted $\ell_1$ distance and Wasserstein duality

For convenience, we first restate the theorem.

**Theorem 5.** *If $n = \Omega(n(\varepsilon)) = \Omega\left(k/(\varepsilon^2 \log k)\right)$ and $\varepsilon \ge n^{-c}$,*

$$\Pr(\ell_1^<(p_\varphi, p) \ge \varepsilon) \le \exp(-\Omega(n^{1/11})).$$

In this section, we relate the estimation of sorted distributions to that of distribution properties through a dual definition of the 1-Wasserstein distance.

Recall that we let $\{p\}$ denote the multiset of probability values of a distribution $p \in \Delta_\mathcal{X}$. The sorted $\ell_1$ distance between two distributions $p, q \in \Delta_\mathcal{X}$ is

$$\ell_1^<(p, q) := \min_{q' \in \Delta_\mathcal{X} : \{q'\} = \{q\}} \|p - q'\|_1,$$

which is invariant under domain-symbol permutations on either $p$ or $q$.

For two distributions $\omega, \nu$ over the unit interval $[0, 1]$, let $\Gamma'_{\omega,\nu}$ be the collection of distributions over $[0, 1] \times [0, 1]$ with marginals $\omega$ and $\nu$ on the first and second factors respectively. The 1-*Wasserstein distance*, also known as the *earth-mover distance*, between $\omega$ and $\nu$ is

$$\mathcal{W}_1(\omega, \nu) := \inf_{\gamma \in \Gamma'_{\omega,\nu}} \mathbb{E}_{(X,Y) \sim \gamma} |X - Y|.$$

Equivalently, let $\mathcal{L}_1$ denote the collection of real functions that are 1-Lipschitz on $[0, 1]$. Through duality, one can also define the 1-Wasserstein distance [54] as

$$\mathcal{W}_1(\omega, \nu) = \sup_{f \in \mathcal{L}_1} \left( \mathbb{E}_{X \sim \omega} f(X) - \mathbb{E}_{Y \sim \nu} f(Y) \right).$$

For any $p \in \Delta_{\mathcal{X}}$, let $u_{\{p\}}$ denote the distribution induced by the uniform measure on $\{p\}$. For any distributions $p, q \in \Delta_{\mathcal{X}}$, one can verify [81, 38, 43] that

$$\ell_1^<(p, q) = k \cdot \mathcal{W}_1(u_{\{p\}}, u_{\{q\}}) \leq R(p, q).$$

Combining this with the dual definition of $\mathcal{W}_1$, we obtain

$$\ell_1^<(p, q) = k \cdot \sup_{f \in \mathcal{L}_1} \left( \mathbb{E}_{X \sim u_{\{p\}}} f(X) - \mathbb{E}_{Y \sim u_{\{q\}}} f(Y) \right) = \sup_{f \in \mathcal{L}_1} \left( \sum_x f(p(x)) - \sum_x f(q(x)) \right).$$

## 7.2 Proof of Theorem 5

For a real function $f \in \mathcal{L}_1$, we denote by $f(p) := \sum_x f(p(x))$ the corresponding additive symmetric property. The previous reasoning also shows that for any $p, q \in \Delta_{\mathcal{X}}$,

$$R(p, q) \geq \ell_1^<(p, q) \geq |f(p) - f(q)|.$$

Therefore, property $f$ is 1-Lipschitz on $(\Delta_{\mathcal{X}}, R)$.

Set $n := \sup_{f \in \mathcal{L}_1} n_f(\varepsilon)$. The results in [43] imply that if $\varepsilon > n^{-0.3}$,

$$n = \Theta \left( \frac{k}{\varepsilon^2 \log k} \right).$$

Clearly, we only need to consider $\varepsilon \leq 2$, implying $k = \mathcal{O}(n \log n)$. Let $\alpha, \gamma$ be absolute constants in $[1/100, 1/6)$ and $\varepsilon > 0$ be an error parameter.

By the proof of Theorem 1 in Section 5.2, for any distribution $p \in \Delta_{\mathcal{X}}$ and $X^{n/\alpha} \sim p$, with probability at least $1 - 2 \exp \left( -4n^{1-2\gamma} \right)$, the PML (or APML) plug-in estimator will satisfy

$$|f(p) - f(p_{\varphi(X^{n/\alpha})})| < \varepsilon (2 + o(1)) + \mathcal{O}(n^{-c_1/2} \log^2 n) + 4n^{1-\gamma} \tau(n),$$

where $c_1 \in (1/100, 1/32]$, $c_2 = 1/2 + 6c_1$, and $\tau(n) = \mathcal{O} \left( n^{\alpha c_2 + (2-\alpha)c_1 - 1} \log^3 n \right)$. Additionally, in the previous section, we have proved that

$$\ell_1^<(p, q) = \sup_{f \in \mathcal{L}_1} (f(p) - f(q)) = \sup_{f \in \mathcal{L}_1} |f(p) - f(q)|.$$

Though it seems that the above inequality and equation imply the optimality of PML (since $f$ is chosen arbitrarily), such direct implication actually does not hold. The reason is a little bit subtle: The inequality on $|f(p) - f(p_{\varphi(X^{n/\alpha})})|$ holds for any fixed function $f$ and $p \in \Delta_{\mathcal{X}}$, while the function that achieves the corresponding supremum in

$$\sup_{f \in \mathcal{L}_1} |f(p) - f(p_{\varphi(X^{n/\alpha})})| = \ell_1^< \left( p, p_{\varphi(X^{n/\alpha})} \right)$$

depends on both $p$ and $X^{n/\alpha}$, and hence is a random function. To address this discrepancy, we provide a more involved argument below.

Let $f$ be a function in $\mathcal{L}_1$. Without loss of generality, we also assume that $f(0) = 0$. Let $\eta \in (0, 1)$ be a threshold parameter to be determined later. An $\eta$-*truncation* of $f$ is a function

$$f_\eta(z) := f(z) \mathbb{1}_{z \leq \eta} + f(\eta) \mathbb{1}_{z > \eta}.$$

One can easily verify that $f_\eta \in \mathcal{L}_1$. Next, we find a finite subset of $\mathcal{L}_1$ so that the $\eta$-truncation of any $f \in \mathcal{L}_1$ is close to at least one of the functions in this subset.

For a parameter $s > 3$ to be chosen later. Partition the interval $[0, \eta]$ into $s$ disjoint sub-intervals of equal length, and define the sequence of end points as $z_j := \eta \cdot j/s, j \in \lceil s \rceil$ where $\lceil s \rceil := \{0, 1, \ldots, s\}$. Then, for each $j \in \lceil s \rceil$, we find the integer $j'$ such that $|f_\eta(z_j) - z_{j'}|$ is minimized and denote it by $j^*$. Since $f_\eta$ is 1-Lipschitz, we must have $|j^*| \in \lceil j \rceil$. Finally, we connect the points $Z_j := (z_j, z_{j^*})$ sequentially. This curve is continuous and corresponds to a particular $\eta$-truncation $\tilde{f}_\eta \in \mathcal{L}_1$, which we refer to as the *discretized $\eta$-truncation* of $f$. Intuitively, we have constructed an $(s+1) \times (s+1)$ grid and "discretized" function $f$ by finding its closest approximation in $\mathcal{L}_1$ whose curve only consists of edges and diagonals of the grid cells. By construction,

$$\max_{z \in [0,1]} |f_\eta(z) - \tilde{f}_\eta(z)| \le \eta/s.$$

Therefore, for any $p \in \Delta_{\mathcal{X}}$, the corresponding properties of $f_\eta$ and $\tilde{f}_\eta$ satisfy

$$|f_\eta(p) - \tilde{f}_\eta(p)| \le k \cdot \eta/s.$$

Note that $|j^*| \in \lceil j \rceil$ for all $j \in \lceil s \rceil$, and $\tilde{f}_\eta(z) = z_{s^*}$ for $z \ge \eta$. While there are infinitely many $\eta$-truncations, the cardinality of the discretized $\eta$-truncations of functions in $\mathcal{L}_1$ is at most

$$\prod_{j=0}^{s}(2j+1) = (s+1)\prod_{j=0}^{s-1}(2j+1)(2s-2j+1) \le (s+1)^{2s+1} = e^{(2s+1)\log(s+1)} \le e^{3s\log s}.$$

Consider any $p \in \Delta_{\mathcal{X}}$ and $X^{n/\alpha} \sim p$ with a profile $\varphi$. Consolidate the previous results, and apply the union bound and triangle inequality. With probability at least $1 - 2\exp\left(3s\log s - 4n^{1-2\gamma}\right)$, the PML plug-in estimator will satisfy

$$|f_\eta(p) - f_\eta(p_\varphi)| \le |f_\eta(p) - \tilde{f}_\eta(p)| + |\tilde{f}_\eta(p) - \tilde{f}_\eta(p_\varphi)| + |\tilde{f}_\eta(p_\varphi) - f_\eta(p_\varphi)|$$
$$\le 2k \cdot \eta/s + \varepsilon\left(2 + o(1)\right) + \mathcal{O}(n^{-c_1/2}\log^2 n) + 4n^{1-\gamma}\tau(n),$$

for *all* functions $f$ in $\mathcal{L}_1$.

Next we consider the "second part" of a function $f \in \mathcal{L}_1$, namely,

$$\bar{f}_\eta(z) := f(z) - f_\eta(z) = (f(z) - f(\eta))\mathbb{1}_{z>\eta}.$$

Again, we can verify that $\bar{f}_\gamma \in \mathcal{L}_1$. To establish the corresponding guarantees, we make use of the following result. Since the profile probability $p(\varphi)$ is invariant to symbol permutation, for our purpose, we can assume that $p(y) \le p(z)$ iff $p_\varphi(x) \le p_\varphi(y)$, for all $x, y \in \mathcal{X}$. Under this assumption, the next lemma, which follows from the consistency results in [67, 8], relates $p_\varphi$ to $p$. Let $\gamma' \in (0, 1/4)$ be an absolute constant to be determined later. Then,

**Lemma 9.** *For any distribution $p$ and sample $X^m \sim p$ with profile $\varphi$,*

$$\Pr\left(\max_x |p_\varphi(x) - p(x)| > m^{\gamma'-1/4}\right) = \mathcal{O}\left(m^{1/4}\exp(-\Omega(m^{1/2+2\gamma'}))\right).$$

Simply following the proofs in [67, 8], we obtain: Changing $1/4$ to any (fixed) number greater than $1/6$, the above lemma also holds for APML with $m^{1/2+2\gamma'}$ replaced by $m^{2/3+2\gamma'}$.

Set $m = n/\alpha$ in this lemma. With probability at least $1 - \mathcal{O}\left((n/\alpha)^{1/4}\exp(-\Omega((n/\alpha)^{1/2+2\gamma'}))\right)$,

$$|\bar{f}_\eta(p) - \bar{f}_\eta(p_\varphi)| = |\sum_x \bar{f}_\eta(p(x)) - \bar{f}_\eta(p_\varphi(x))|$$
$$\le \sum_{x:p(x)>\eta \text{ or } p_\varphi(x)>\eta} |\bar{f}_\eta(p(x)) - \bar{f}_\eta(p_\varphi(x))|$$
$$\le \sum_{x:p(x)>\eta \text{ or } p_\varphi(x)>\eta} |p(x) - p_\varphi(x)|$$
$$\le (2/\eta)(n/\alpha)^{\gamma'-1/4},$$

for *all* functions $f$ in $\mathcal{L}_1$.

Consolidate the previous results. By the triangle inequality and the union bound, with probability at least $1 - 2\exp\left(3s\log s - 4n^{1-2\gamma}\right) - \mathcal{O}\left((n/\alpha)^{1/4}\exp(-\Omega((n/\alpha)^{1/2+2\gamma'}))\right)$,

$$|f(p) - f(p_\varphi)| \leq |f_\eta(p) - f_\eta(p_\varphi)| + |\bar{f}_\eta(p) - \bar{f}_\eta(p_\varphi)|$$
$$\leq 2k\eta/s + \varepsilon\left(2 + o(1)\right) + \mathcal{O}(n^{-c_1/2}\log^2 n) + 4n^{1-\gamma}\tau(n) + (2/\eta)(n/\alpha)^{\gamma'-1/4},$$

for *all* functions $f$ in $\mathcal{L}_1$. Now we can conclude that $\ell_1^<(p, p_\varphi)$ is also at most the error bound on the right-hand side. The reason is straightforward: Since with high probability, the above guarantee holds for all functions in $\mathcal{L}_1$, it must also hold for the function that achieves the supremum in

$$\sup_{f\in\mathcal{L}_1} |f(p) - f(p_\varphi)| = \ell_1^<(p, p_\varphi).$$

It remains to make sure that all the quantities in the error bound except $\varepsilon\left(2 + o(1)\right)$ vanish with $n$, and the probability bound converges to 1 as $n$ increases. Recall that $k = \mathcal{O}(n\log n)$, $c_1 \in (1/100, 1/25]$, $c_2 = 1/2 + 6c_1$, and $\tau(n) = \mathcal{O}\left(n^{\alpha c_2 + (2-\alpha)c_1 - 1}\log^3 n\right)$.

By direct computation, we can choose $\alpha = 1/100$, $c_1 = 1/26$, $\gamma' = 1/200$, $\gamma = (5/2+5\alpha)c_1 + \alpha/2$, $s = n^{\gamma'+3/4+c_1}$, and $\eta = n^{\gamma'-1/4+c_1/2}$. Note that this is just one possible set of parameters. Given this choice, we have

$$\ell_1^<(p, p_\varphi) \leq \varepsilon\left(2 + o(1)\right) + \mathcal{O}(n^{-c_1/2}\log^3 n),$$

with probability at least $1 - \exp(-\Omega(n^{1/2}))$. Additionally, the equation

$$\sup_{f\in\mathcal{L}_1} |f(p) - f(p_\varphi)| = \ell_1^<(p, p_\varphi)$$

clearly yields that $n(\varepsilon) \geq \sup_{f\in\mathcal{L}_1} n_f(\varepsilon)$. Hence for $\varepsilon \geq \mathcal{O}(n^{-c_1/2}\log^4 n)$,

$$n(p_\varphi, (2 + o(1))\varepsilon) \leq 100n(\varepsilon).$$

# 8 Uniformity testing

## 8.1 PML-based tester

Let $\varepsilon$ be an arbitrary accuracy parameter and $\mathcal{X}$ be a finite set. Let $p_u$ denote the uniform distribution over $\mathcal{X}$. Given sample access to an unknown distribution $p \in \Delta_\mathcal{X}$, the uniformity testing distinguishes between the null hypothesis

$$H_0 : p = p_u$$

and the alternative hypothesis

$$H_1 : \|p - p_u\|_1 \geq \varepsilon.$$

After a sequence of research works [37, 13, 71, 3, 19, 78, 31, 4, 30, 32], it is shown that to achieve a $k^{-\Theta(1)}$ bound on the error probability, this task requires a worst-case sample size of order $\sqrt{k\log k}/\varepsilon^2$. The uniformity tester $T_{\text{PML}}(X^n)$ in Figure 7 is purely based on PML, and takes as input parameters $k$ and $\varepsilon$, and a sample $X^n \sim p$.

---

**Input:** parameters $k, \varepsilon$, and a sample $X^n \sim p$ with profile $\varphi$.
1. If $\max_x \mu_x(X^n) \geq 3\max\{1, n/k\}\log k$, return 1;
2. Elif $\|p_\varphi - p_u\|_2 \geq 3\varepsilon/(4\sqrt{k})$, return 1;
3. Else return 0.

---

Figure 7: Uniformity tester $T_{\text{PML}}$

In the rest of this section, we establish the following theorem.

**Theorem 6.** *If $\varepsilon = \tilde{\Omega}(k^{-1/4})$ and $n = \tilde{\Omega}(\sqrt{k}/\varepsilon^2)$, then the tester $T_{PML}(X^n)$ will be correct with probability at least $1 - k^{-2}$. The tester also distinguishes between $p = p_u$ and $\|p - p_u\|_2 \geq \varepsilon/\sqrt{k}$.*

## 8.2 Proof of Theorem 6

Assume that $\varepsilon \geq (\log k)/k^{1/4}$. For a sample $X^n \sim p_u$, the multiplicity of each symbol $x$ follows a binomial distribution $\text{bin}(n, k^{-1})$ with mean $n/k$. The following lemma [25] bounds the tail probability of a binomial random variable.

**Lemma 10.** *For a binomial random variable $Y$ with mean $M$ and any $t \geq 1$,*
$$\Pr(Y \geq (1+t)M) \leq \exp(-t(2/t + 2/3)^{-1}M).$$

Applying the above lemma to $Y = \mu_x(X^n)$ and $t = 3\max\{k/n, 1\}\log k$ immediately yields that $\Pr(\mu_x(X^n) \geq (1+t)n/k) \leq k^{-3}$. By symmetry and the union bound, we then have $\Pr(\max_x \mu_x(X^n) \geq (1+t)n/k) \leq k^{-2}$. In the subsequent discussion, we denote by $\Phi_{\mathcal{X}}^n$ the profile set $\{\varphi(x^n) : x^n \in \mathcal{X}^n \text{ and } \max_x \mu_x(x^n) < (1+t)n/k\}$.

Consider the problem of estimating the $\ell_2$-distance between an unknown distribution and the uniform distribution $p_u$, for which we have the following result [36].

**Lemma 11.** *There is a profile-based estimator $\hat{\ell}_2$ such that for any $\varepsilon_0 \leq k^{-1/2}$, $n = \Omega(k^{-1/2}/\varepsilon_0^2)$, $p \in \Delta_{\mathcal{X}}$ satisfying $P_2(p) = \mathcal{O}(k^{-1})$, and $X^n \sim p$,*

- *if $\|p - p_u\|_2 > \varepsilon_0$, then $\hat{\ell}_2(X^n) \geq 0.9\varepsilon_0$,*
- *if $\|p - p_u\|_2 < \varepsilon_0/2$, then $\hat{\ell}_2(X^n) \leq 0.6\varepsilon_0$,*

*with probability at least $2/3$.*

Set $\varepsilon_0 = \varepsilon/\sqrt{k}$ in the above lemma. Then, by the sufficiency of profiles and the standard median trick, there exists another profile-based estimator $\hat{\ell}_2^\star$ that under the same conditions, provides the estimation guarantees stated above, with probability at least $1 - \delta$ for $\delta := 2\exp(-\Omega(n\varepsilon^2/\sqrt{k}))$. Scaling $\varepsilon_0$ by positive absolute constant factors yields: If $\|p - p_u\|_2 > 0.67\varepsilon_0$, then $\hat{\ell}_2(X^n) \leq 0.6\varepsilon_0$ with probability at most $\delta$; if $\|p - p_u\|_2 < 0.75\varepsilon_0$, then $\hat{\ell}_2(X^n) \geq 0.9\varepsilon_0$ with probability at most $\delta$.

Let $\varphi'$ be a profile. If we further have $p(\varphi') > \delta$, then by definition, $p_{\varphi'}(\varphi') \geq p(\varphi') > \delta$. Hence for any $x^n$ with profile $\varphi'$, if $\|p - p_u\|_2 > \varepsilon_0$, we must have both $\hat{\ell}_2(x^n) \geq 0.9\varepsilon_0$ and $\|p_{\varphi'} - p_u\|_2 \geq 0.75\varepsilon_0$; if $\|p - p_u\|_2 < \varepsilon_0/2$, we must have both $\hat{\ell}_2(x^n) \leq 0.6\varepsilon_0$ and $\|p_{\varphi'} - p_u\|_2 \leq 0.67\varepsilon_0$.

On the other hand, for a sample $X^n \sim p$ with profile $\varphi$, the probability that we have both $p(\varphi) \leq \delta$ and $\varphi \in \Phi_{\mathcal{X}}^n$ is at most $\delta$ times the cardinality of the set $\Phi_{\mathcal{X}}^n$. By definition, if $\varphi \in \Phi_{\mathcal{X}}^n$, then $\varphi_i = 0$ for $i \geq (1+t)n/k$. In addition, each $\varphi_i$ can only take values in $\lceil k \rceil = \{0, 1, \ldots, k\}$, implying that $|\Phi_{\mathcal{X}}^n| \leq |\lceil k \rceil|^{(1+t)n/k} \leq \exp(6\max\{n/k, 1\}\log^2 k)$. Therefore, we obtain the following upper bound on the probability of interest: $\delta \cdot |\Phi_{\mathcal{X}}^n| \leq 2\exp(-\Omega(n\varepsilon^2/\sqrt{k}) + 6\max\{n/k, 1\}\log^2 k)$. In order to make the probability bound vanish, we need to consider two cases: $n \leq k$ and $n > k$. If $n \leq k$, it suffices to have $n \gg (\log^2 k)\sqrt{k}/\varepsilon^2$; If $n > k$, it suffices to have $\varepsilon \gg (\log k)/k^{1/4}$. In both cases, the probability bound is at most $\exp(-\log^2 k)$.

Next, consider estimating the power sum $P_2(p)$, which is at least $k^{-1/2}$ for $p \in \Delta_{\mathcal{X}}$. By Corollary 2, there is a profile-based estimator $\hat{P}_2^\star$ such that $\Pr_{X^n \sim p}(|\hat{P}_2^\star(X^n) - P_2(p)| \geq (\varepsilon/8) \cdot P_2(p)) \leq 2\exp(-\Omega(n\varepsilon^2/\sqrt{k})) = \delta$. Following the same derivations as above and in Section 6.3 with $\Phi_{\alpha,\varepsilon}^n(p)$ replaced by $\Phi_{\mathcal{X}}^n$, we establish that
$$\Pr(|P_2(p_\varphi) - P_2(p)| > P_2(p)/2 \text{ and } \varphi \in \Phi_{\mathcal{X}}^n) \leq \delta \cdot |\Phi_{\mathcal{X}}^n| \leq \exp(-\log^2 k).$$

Now we are ready to characterize the performance of the tester $T_{\text{PML}}(X^n)$. For clarity, we divide our analysis into two parts based on which hypothesis is true.

- **Case 1:** The null hypothesis $H_0$ is true, i.e., $p = p_u$.
  - **Step 1:** By Lemma 10 and its implications, given $p = p_u$, the probability of failure at this step is at most $\Pr_{X^n \sim p_u}(\exists x \in \mathcal{X} \text{ s.t. } \mu_x(X^n) \geq (1+t)n/k) \leq k^{-2}$.
  - **Step 2:** Note that $P_2(p) = k^{-1}$ and $\|p - p_u\|_2 = 0$, and recall that $\varphi = \varphi(X^n)$. The tester accepts $H_1$ in this step iff $\varphi \in \Phi_{\mathcal{X}}^n$ and $\|p_\varphi - p_u\|_2 \geq 0.75\varepsilon_0$. By Lemma 11 and the subsequent arguments, this happens with probability at most $\exp(-\log^2 k)$.

- **Step 3:** The tester always accepts $H_0$ in this step. Hence by the union bound, if the null hypothesis $H_0$ is true, then the tester succeeds with probability at least $1 - k^{-2}$.

- **Case 2:** The alternative hypothesis $H_1$ is true, i.e., $\|p - p_u\|_1 \geq \varepsilon$.

  - **Step 1 to 2:** The tester accepts $H_1$ if the conditions in either Step 1 or Step 2 are satisfied, and hence incurs no error.

  - **Step 3:** According to the value of $P_2(p)$, we further divide our analysis into two parts:

    * If $P_2(p) \geq 10k^{-1}$, then $\|p_\varphi - p_u\|_2 < 0.75\varepsilon/\sqrt{k}$ implies that $P_2(p_\varphi) < 1.6k^{-1}$ and $|P_2(p_\varphi) - P_2(p)| > P_2(p)/2$. Hence, the tester accepts $H_0$ only if both $|P_2(p_\varphi) - P_2(p)| > P_2(p)/2$ and $\varphi \in \Phi^n_{\mathcal{X}}$ happen, whose probability, by the above disscusion, is at most $\exp(-\log^2 k)$.

    * If $P_2(p) < 10k^{-1}$, then all the conditions in Lemma 11 are satisfied. In addition, by the Cauchy-Schwarz inequality, we have $\|p - p_u\|_2 \geq \|p - p_u\|_1 \cdot k^{-1/2} \geq \varepsilon \cdot k^{-1/2}$. The tester accepts $H_0$ iff both $\|p_\varphi - p_u\|_2 < 0.75\varepsilon \cdot k^{-1/2}$ and $\varphi \in \Phi^n_{\mathcal{X}}$ hold, which happen, by Lemma 11 and the subsequent arguments, with probability at most $\exp(-\log^2 k)$.

This completes the proof of the theorem.

## Conclusion

We studied three fundamental problems in statistical learning: distribution estimation, property estimation, and property testing. We established the profile maximum likelihood (PML) as the first universally sample-optimal approach for several important learning tasks: distribution estimation under the sorted $\ell_1$ distance, additive property estimation, Rényi entropy estimation, and identity testing. Several future directions are promising. We believe that neither the factor of $4$ in the sample size in Theorem 1, nor the lower bounds on $\varepsilon$ in Theorem 1, 5, and 6 are necessary. In other words, the PML approach is universally sample-optimal for these tasks in all ranges of parameters. It is also of interest to extend the PML's optimality to estimating symmetric properties not covered by Theorem 1 to 4, such as *generalized distance to uniformity* [10, 44], the $\ell_1$ distance between the unknown distribution and the closest uniform distribution over an arbitrary subset of $\mathcal{X}$.

Another important direction is *competitive (or instance-optimal) property estimation*. It should be noted that all the referenced works including this paper are of the worst-case nature, namely, designing estimators with near-optimal worst-case performances. On the contrary, practical and natural distributions often possess simple structures, and are rarely the worst possible. To address this discrepancy, the recent work [45, 48] took a competitive approach and constructed estimators whose performances are adaptive to the simplicity of the underlying distributions. Specifically, for any property in a broad class and *every* distribution in $\Delta_{\mathcal{X}}$, the expected error of the proposed estimator with a sample of size $n/\log n$ is at most that of the empirical estimator with a sample of size $n$, pluses a distribution-free vanishing function of $n$. These results not only cover $\tilde{S}$, $\tilde{C}_m$, $H$, and $D$, for which the $\log n$-factor is optimal up to constants, but also apply to any *non-symmetric additive* property $\sum_x f_x(p_x)$ where $f_x$ is 1-Lipschitz for all $x \in \mathcal{X}$, such as the $\ell_1$-distance to a given distribution. It would be of interest to study the optimality of the PML approach under this formulation as well. Readers interested in estimating non-symmetric properties may also find the paper [47] helpful.

## A   Proof of Lemma 3

The proof closely follows that of Proposition 6.19 in [83] (page 131–136), which we refer to as *the proposition's proof.* Note that in the work [83], the definitions of $k$ and $n$ are swapped, i.e., $k$ stands for the sample size, and $n$ denotes the alphabet size. For consistency, we still keep our notation.

Recall that we set $t_n := 2n^{-c_1}\log n$ and $\alpha \in (0,1)$, and define

$$\beta_i := (1 - e^{-t_n \alpha i})f\left(\frac{(i+1)\alpha}{n}\right)\frac{n}{(i+1)\alpha} + \sum_{\ell=0}^{i} z_\ell(1-t_n)^\ell \alpha^\ell (1-\alpha)^{i-\ell}\binom{i}{\ell}.$$

for any $i \leq n$, and $\beta_i := \beta_n$ for $i > n$. Let $w(i)$ denote the first quantity on the right-hand side, and $w := (w(0), w(1), \ldots)$ be the corresponding vector. Similarly, let $\tilde{z}_\alpha(i)$ denote the second quantity on the right-hand side, and $\tilde{z}_\alpha$ be the corresponding vector. Assume that $v \leq \log^2 n$.

First part of the proposition's proof remains unchanged, which corresponds to the content from page 131 to the second last paragraph on page 132, showing that

$$\sqrt{\alpha} \, \|\tilde{z}_\alpha\|_2 = \mathcal{O}(n^{\alpha c_2 + (1-\alpha)c_1} \cdot \log^3 n).$$

The assumption that $\alpha \in [1/100, 1)$ implies $\sqrt{\alpha} \geq 1/10$, and hence we have $|\tilde{z}_\alpha(i)| \leq \|\tilde{z}_\alpha\|_2 = \mathcal{O}(n^{\alpha c_2 + (1-\alpha)c_1} \cdot \log^3 n)$. Recall that for lemma 2 to hold, the coefficients $\beta_i$ must satisfy the following two conditions,

1. $|\varepsilon(y)| \leq a' + b'/y$,
2. $|\beta_j^\star - \beta_\ell^\star| \leq c' \sqrt{j/n}$ for any $j$ and $\ell$ such that $|j - \ell| \leq \sqrt{j} \log n$,

where $\varepsilon(y) := f(y)/y - e^{-ny} \sum_{i \geq 0} \beta_i \cdot (ny)^i/i!$, and $\beta_i^\star := \beta_{i-1} \cdot i/n, \forall i \geq 1$, and $\beta_0^\star := 0$.

We first consider the second condition and find a proper parameter $c'$.

Our objective is to find $c' > 0$ such that $c' > \sqrt{n/j} \, |\beta_j^\star - \beta_\ell^\star|$. By the triangle inequality,

$$\sqrt{\frac{n}{j}} |\beta_j^\star - \beta_\ell^\star| \leq \sqrt{\frac{n}{j}} \left| \frac{j}{n} \tilde{z}_\alpha(j-1) - \frac{\ell}{n} \tilde{z}_\alpha(\ell-1) \right| + \sqrt{\frac{n}{j}} \left| \frac{j}{n} w(j-1) - \frac{\ell}{n} w(\ell-1) \right|$$

We bound the two quantities on the right-hand side separately and consider two cases for each. If both $j$ and $\ell$ are at most $400 n^{c_1}$, then

$$\sqrt{\frac{n}{j}} \left| \frac{j}{n} \tilde{z}_\alpha(j-1) - \frac{\ell}{n} \tilde{z}_\alpha(\ell-1) \right| \leq \mathcal{O}(n^{c_1/2 - 1/2}) \cdot \max_i |z_\alpha(i)| \leq \mathcal{O}(n^{\alpha c_2 + (3/2 - \alpha)c_1 - 1/2} \log^3 n).$$

Recall that $|z_\ell| \leq v \cdot n^{c_2}, \forall \ell \geq 0$. If one of $j$ and $\ell$ is larger than $400 n^{c_1}$, say $j > 400 n^{c_1}$, then

$$\sqrt{\frac{n}{j}} \left| \frac{j}{n} \tilde{z}_\alpha(j-1) \right| \leq \sqrt{\frac{j}{n}} \sum_{\ell=0}^{j-1} |z_\ell| (1-t_n)^\ell \alpha^\ell (1-\alpha)^{j-1-\ell} \binom{j-1}{\ell}$$

$$\leq \sqrt{j} n^{c_2 - 1/2} (\log^2 n) \sum_{\ell=0}^{j-1} (1-t_n)^\ell \alpha^\ell (1-\alpha)^{j-1-\ell} \binom{j-1}{\ell}$$

$$= \sqrt{j} n^{c_2 - 1/2} (\log^2 n)(1 - t_n \alpha)^{j-1}$$

$$\leq \sqrt{j} n^{c_2 - 1/2} (\log^2 n)(1 - \log n/(50 n^{c_1}))^{400 n^{c_1}}$$

$$\leq \sqrt{j} n^{c_2 - 1/2} (\log^2 n) n^{-8}.$$

For $j < 2n^2$, the last quantity is at most $n^{-1}$. For $j > 2n^2$, we have $\ell > n^2$ and hence

$$\sqrt{\frac{n}{j}} \left| \frac{j}{n} \tilde{z}_\alpha(j-1) - \frac{\ell}{n} \tilde{z}_\alpha(\ell-1) \right| = \sqrt{\frac{n}{j}} \, |j - \ell| \, \tilde{z}_\alpha(n-1) \leq \sqrt{n} (\log n) n^{-1} = (\log n) n^{-1/2}.$$

Similarly, we can bound the other quantity, i.e.,

$$\sqrt{\frac{n}{j}} \left| \frac{j}{n} w(j-1) - \frac{\ell}{n} w(\ell-1) \right| = \sqrt{\frac{n}{\alpha^2 j}} \left| (1 - e^{-t_n \alpha(j-1)}) f\left(\frac{j\alpha}{n}\right) - (1 - e^{-t_n \alpha(\ell-1)}) f\left(\frac{\ell\alpha}{n}\right) \right|.$$

Since $f$ (the property) is 1-Lipschitz on $(\Delta_\mathcal{X}, R)$ and $f(p) = 0$ if $p(x) = 1$ for some $x \in \mathcal{X}$, one can verify that $|f(x)| \leq x |\log x| \leq e^{-1}$ and $|f(x)/x - f(y)/y| \leq |\log(x/y)|$ for $x, y \in [0, 1]$ (the corresponding real function). We consider two cases and bound the quantity of interest. If $j \geq \sqrt{n}$,

$$\sqrt{\frac{n}{\alpha^2 j}} \left| (1 - e^{-t_n \alpha(j-1)}) f\left(\frac{j\alpha}{n}\right) \right| \leq \sqrt{\frac{n}{\alpha^2 j}} \left| f\left(\frac{j\alpha}{n}\right) \right| \leq \sqrt{\frac{n}{\alpha^2 j}} \frac{j\alpha}{n} \log\left(\frac{j\alpha}{n}\right) \leq \mathcal{O}(n^{-1/4} \log n).$$

The same bound also applies to the other term where $j$ is replaced by $\ell$. If $j > \sqrt{n}$, then $e^{-t_n \alpha (j-1)} \leq \exp\left(-2\alpha(\log n)n^{1/2-c_1}\right) = \mathcal{O}(n^{-2})$. Analogously, the same upper bound holds for the other term $e^{-t_n \alpha(\ell-1)}$. Hence, we ignore these two terms and consider only

$$
\begin{aligned}
\sqrt{\frac{n}{\alpha^2 j}}\left|f\left(\frac{j\alpha}{n}\right) - f\left(\frac{\ell\alpha}{n}\right)\right| &\leq \sqrt{\frac{j}{n}}\left|\frac{n}{j\alpha}f\left(\frac{j\alpha}{n}\right) - \frac{n}{\ell\alpha}f\left(\frac{\ell\alpha}{n}\right)\right| + \sqrt{\frac{j}{n}}\left|\frac{n}{j\alpha} - \frac{n}{\ell\alpha}\right|f\left(\frac{\ell\alpha}{n}\right) \\
&\leq \sqrt{\frac{j}{n}}\left|\log\frac{j}{\ell}\right| + \sqrt{\frac{j}{n}}\left|\frac{n}{j\alpha} - \frac{n}{\ell\alpha}\right|f\left(\frac{\ell\alpha}{n}\right) \\
&\leq \sqrt{\frac{j}{n}}\frac{|j-\ell|}{j} + \frac{\sqrt{jn}}{\alpha}\frac{|j-\ell|}{j\ell}f\left(\frac{\ell\alpha}{n}\right) \\
&\leq \sqrt{\frac{j}{n}}\frac{|j-\ell|}{j} + \sqrt{\frac{j}{n}}\frac{|j-\ell|}{j}\left|\log\left(\frac{\ell\alpha}{n}\right)\right| \\
&\leq \frac{\log n}{\sqrt{n}} + \frac{\log n}{n}\left|\log\left(\frac{\ell\alpha}{n}\right)\right| \\
&= \mathcal{O}(n^{-1/2}\log n).
\end{aligned}
$$

By the assumption that $\alpha c_2 + (3/2 - \alpha)c_1 \leq 1/4$, we have $\mathcal{O}(n^{\alpha c_2 + (3/2 - \alpha)c_1 - 1/2}\log^3 n) = \mathcal{O}(n^{-1/4}\log^3 n)$. Hence, we can set the latter quantity to be $c'$. The above derivations also show that

$$
|w(i)| = \left|(1 - e^{-t_n\alpha i})f\left(\frac{(i+1)\alpha}{n}\right)\frac{n}{(i+1)\alpha}\right| \leq \left|\log\left(\frac{(i+1)\alpha}{n}\right)\right| = \mathcal{O}(\log n).
$$

Together with $\beta_i = w(i) + \tilde{z}_\alpha(i)$ and $|\tilde{z}_\alpha(i)| = \mathcal{O}(n^{\alpha c_2 + (1-\alpha)c_1} \cdot \log^3 n)$, this inequality implies

$$
|\beta_i| \leq \mathcal{O}(n^{\alpha c_2 + (1-\alpha)c_1}\log^3 n).
$$

It remains to analyze the first condition of Lemma 2 and find proper values for $a'$ and $b'$. For this part, the corresponding proof in [83] also holds for $\alpha \in [1/100, 1/2]$ (page 134 to the second last paragraph on page 135), hence no change is needed. One thing to note is that $1/\alpha$ and $1/\sqrt{\alpha}$ are both $\mathcal{O}(1)$. For some $a'', b'' \geq 0$ such that $a'' + b''k \leq v$, we can set $a' = a'' + \mathcal{O}(n^{-c_1/2}\log^2 n)$ and $b' = b''(1 + \mathcal{O}(n^{-c_1}\log n))$. The proof of Lemma 3 is complete.