[Reviews · NeurIPS 2019]

Reviewer 1



Originality: While the PML estimator was proposed in prior work and analyzed in a series of papers, this paper is the first to analyze it in such generality. Results for estimating specific properties also seem to be state-of-the-art, and connections to prior work are well discussed. Clarity and quality: The exposition is clear, and while I did not check all proofs in detail, the ones I did check seem to be correct. Significance: The PML estimator is a natural estimator for these problems, and providing a unified view of its properties is significant. This work also raises interesting algorithmic questions regarding the approximate PML that could be the subject of further research.

Reviewer 2



Following work of Acharya,, Das, Orlitsky, and Suresh [5] which showed that a single estimator, the "profile maximum likelihood" (PML) and its approximate version (APML) achieve near-sample optimal guarantees for several property estimation tasks, this work investigates the applicability of the (A)PML for testing and estimation tasks on discrete distributions: - Sorted distribution estimation: estimate the vector of sorted frequencies - symmetric property estimation: estimating the Shannon, Renyi entropy of a distribution, or its support size, etc. - identity testing: (one-sample testing) This work shows that the (A)PML is near-optimal (in terms of sample size) for the above tasks, up to either constants or polylogarithmic factors (in the domain size k). ================= Comments: - The organization of the introduction seems strange to me. It starts by introducing additive symmetric properties before anything else, and devotes a lot of space to it; however, it then pivots to "problems of interest" of which two are not additive symmetric properties... this needs reorganizing to make more sense, outline-wise. - l.50 (and abstract): do not title this "distribution estimation": write "sorted frequencies estimation", or "sorted distribution estimation" - the problem you are solving is *not* the standard distribution estimation one. - abstract and l.49 and throughout: "statistical-learning" -> "statistical learning"; l.70: "median-trick" -> "median trick" - l.104, l.285: why do you have *three* citations for the Charikar et al. paper (one talk, one arxiv, one peer-reviewed)? It's one paper, cite it as one paper. ([23], *not* [21,22,23]) l.115: no need to cite [21] again (also, that'd be [23]): you mentioned that near-linear computation a few lines before. l. 123: log|eps| is somewhat strange. log(1/eps) is more readable and intuitive. ***l.130: I do not understand. How can it be that an *approximate* (hence, "not as good") computation of the PML can give *better* bounds than the PML? Is there a mistake? Can you explain this? ***Theorem 6: What about constant error probability? Can you achieve the optimal sqrt(k)/eps^2 for that? - l.165: you are being a bit misleading here, as *nearly all* uniformity testers (except Paninski'08 and [31]) actually provide this very same L2 testing guarantee (as they basically work by counting collisions, or a chi^2 variant thereof) - l.251: actually, maybe give a short motivation for this sorted L1 distance estimation? You never did, and it does have applications, so... l. 271: [3] is not relevant (it's not the same testing setting, and it's under monotonicity assumptions), and [17] either (it's 2-sample testing (closeness), not identity; the l2 tester they give is relevant, but then, never phrased in terms of identity testing, and basically subsumed by other works anyway) === Question: you mention several times PML as a "universal tester" (or property estimator). However, can you get a *truly* universal statement in terms of quantification (and error probability)? "with probability 9/10, the APML computed is (gives tests that are) optimal *simultaneously* for all the following tasks: [...]" (the way that a plugin approach with the empirical estimator would work, given O(k/eps^2) samples: as long as the estimate is close to the true distribution in TV distance, *all* plugin estimators computed on it are accurate at the same time) UPDATE: I read the authors' response, and am satisfied with it.

Reviewer 3



The paper considers various kinds of canonical statistical inference problems which aim to infer properties of an unknown discrete distribution p based on n independent samples from it. The considered problems include property estimation where the goal is to estimate f(p) up to certain accuracy for some function f, distribution estimation where the goal is to estimate the true distribution up to a certain distance, and property testing, where the goal is to determine whether p is close to a known distribution q or not. These problems have been studied since decades ago and received a lot of interest in both statistics and machine learning community recently in the finite sample regime. Sample optimal estimators have been proposed seperately for different problems recently. This paper approaches these problems based on a universal plug-in estimator using profile maximum likelihood (PML) estimation for the unknown distribution p, which outputs a distribution which maximizes the probability of outputing the profile (histogram of histograms) of the sample sequence. They showed that: 1) The estimator achieves optimal sample complexity for sorted distribution estimation for a wide range of parameters. It also improves upon the dependence on error probability over previous works. This also holds for approximate PML (APML), which can be computed in near-linear time. 2) The estimator achieves optimal sample complexity up to constant factors for all additive symmetric properties that are 1-Lipschitz with respect to relative earth mover distance, which include Shannon entropy, Renyi entropy, support size, support coverage, distance to uniformity. This also holds for APML. 3) For Renyi entropy estimation, the estimator also matches the current state of the art for non-integer \alpha's, which is optimal up to log factors. It also improves upon state-of-the-art on its dependence on error probability. For integer \alpha's, it achieves optimal dependence on the alphabet size while the dependence on the error parameter is not optimal. The result also holds for APML. 4) For identity testing, an algorithm based on PML estimator can achieve the optimal sample complexity up to log factors. But it is not clear whether this will hold for its approximate version. The results in this paper are pretty impressive. As this is the first result which shows the optimality of a single plug-in estimator for a wide range of statistical inference problems. The optimality of PML has been shown before for only certain property estimation problems based on a case-by-case study. This paper extends it to estimating a more general class of 1-Lipschitz additive symmetric properties, estimating Renyi entropy and sorted distribution estimation. The paper also demonstrates the efficacy of its proposed estimator on distribution estimation and showed it outperforms the current state-of-the-art Good-Turing estimator. The paper is also nicely written with clearly stated results with detailed comparison to previous results. Cons: 1. The result on property testing is less impressive since it is neither sample optimal nor near linear-time computable. Various computationally efficient and sample optimal estimators have been proposed before. 2. The experiments are based on MCMC-EM algorithm to compute the PML estimation. It would be nice to see how would the near-linear time approximation of PML would work empirically. 3. The experiments for Shannon entropy estimation in the appendix is not impressive. 4. Detailed comments: 1) The notation for the equation between line 78-79 is not clear. A better way might be \forall p \in set H_i. 2) Theorem 1: it might be better to state all the conditions in the theorem to make it more self-contained. 3) Line 104: APML is not properly defined. There might be confusions on what does it mean by an approximation of PML. Overall, I think the paper is among the top papers for NeurIPS. I would strongly recommend this paper to be accepted.

[Author Response · NeurIPS 2019]

We thank all the reviewers for liking our paper and providing positive and insightful feedback. Below we address the
comments and questions regarding our results and writing style.

**Reviewer 1:**  Thank you again for your encouraging comments on the theoretical and experimental results. We too
are happy that a single method could perform really well on a variety of learning tasks. We completely agree that
approximating PML is an interesting and important problem for further research, and will look into it in the near future.

**Reviewer 2:**  We really appreciate your thorough and insightful comments. We have incorporated all of them in our
draft and will submit the new version if the paper gets accepted. Below we present our detailed responses in order.

– Thank you for pointing out that the introduction section put much of its emphasis on the property estimation problem.
We are modifying and reorganizing the introduction to improve its presentation. We will also motivate the other two
learning tasks: sorted distribution estimation and property testing.

– L.50 (and abstract): We have changed "distribution estimation" to "sorted distribution estimation";

– Abstract and L.49 and throughout; L.70: We have removed the hyphens in "statistical-learning" and "median-trick";

– L.104, L.285: We have modified the reference list and cited the Charikar et al. paper as a single paper [23]. In the
submitted version, we had three different citations because the STOC camera-ready version was not available at that
time. Since the result was relatively new, we also included the talk to help potential reviewers better understand it.

– L.115 and L.123: We have removed citation [21] and modified $\log |\epsilon|$ to $\log(1/\epsilon)$.

*** L.130: The constant $c$ used for APML is actually (slightly) worse than that for PML. On the other hand, this makes
it possible to strengthen the error probability bound. We have updated our draft to clarify this.

*** Theorem 6: The current proof does not yield the $\sqrt{k}/\varepsilon^2$ complexity of uniformity testing in the constant confidence
regime. We do have an alternative argument that utilizes the problem structure to achieve the $\sqrt{k}/\varepsilon^2$ sample complexity.
We will provide a sketch of the alternative argument in the updated version.

– L.165: The emphasis here is that our tester is "the first PML-based uniformity tester" providing both the $\ell_1$ and $\ell_2$
testing guarantees. Incorporating your comments, we have modified the statement and pointed out that "nearly all
uniformity testers in the literature [. . .] provide the same $\ell_2$ testing guarantee".

– L.251: We have added a short motivation for the sorted $\ell_1$ distance estimation. There are several motivations, including
but not limited to (unsorted) distribution estimation [70] and symmetric property estimation.

– L.271: We have removed [3] and [17]. We appreciate the detailed comments regarding the references.

"Question": A simple definition of a *universal plug-in property estimator* could be a sentence similar to the one in the
abstract. For example, "there exist absolute positive constants $c_1, c_2$ and $c_3$, such that for any 1-Lipschitz property on
$(\Delta_{\mathcal{X}}, R)$, with probability $\geq 9/10$, the plug-in estimator uses just $c_1$ times the sample size $n$ required by the minimax
estimator to achieve $c_2$ times its error, whenever this error is at least $n^{-c_3}$". We are still thinking about better definitions.

**Reviewer 3:**  Thank you for the encouraging comments. The concise and detailed summary of the paper's contributions
you provided is valuable for us to improve its presentation and organization. We are also excited about the broad
optimality of PML and APML as well as your strong recommendation for the paper's acceptance. Below we mainly
address the "Cons" mentioned in your comments.

1. We agree that the property testing result is not as impressive as the paper's other results. Yet we still find it interesting
because combined with other results, it shows that PML is a generic tool for a variety of inference tasks. In addition,
the $\exp(-3\sqrt{n})$ error probability bound does not hold for property testing [31], hence the "competitiveness arguments"
in [5] do not directly apply here. Instead, we utilize a new concept of "typical profiles" to reduce the number of objects
considered, which in turn requires only weaker concentration. This technique may be of independent interest.

2. Thank you for the nice suggestion of performing experiments using APML, the near-linear computable variant of
PML. The code is not publicly available yet, and we will ask the authors of [23] for the code so we can compare APML
and the MCMC-EM algorithm. We are looking forward to seeing the experimental results.

3. The experiments for Shannon entropy estimation basically showed that the MCMC-EM PML computation algorithm
is as good as state-of-the-art algorithms specifically designed for entropy estimation. One way to improve the estimation
accuracy is to increase the number of EM iterations, e.g., from 30 to 50. On the other hand, this will make the algorithm
around two times slower, since the computation time of each MCMC-EM iteration is roughly the same.

4. Detailed comments: 1) We have modified the notation used in line 78-79 according to your suggestion; 2) We have
simplified the statements of conditions and made Theorem 1 self-contained; 3) We have added a new paragraph to give
a high-level description of the APML algorithm. Thank you for helping us enhance the writing of the paper.

[Meta-Review · NeurIPS 2019]

The paper shows that profile maximum likelihood, an idea from the distribution estimation literature from a couple of years ago, enjoys optimality properties for a large class of property estimation tasks. The class of tasks includes a number of popular problems studied in the distribution learning literature. All reviewers liked the paper and advocate acceptance. Please do go over the reviews and incorporate any feedback for the camera ready.